# *Armc5* deletion causes developmental defects and compromises T-cell immune responses

Yan Hu[1,*], Linjiang Lao[1,*], Jianning Mao[1], Wei Jin[1], Hongyu Luo[1], Tania Charpentier[2], Shijie Qi[1], Junzheng Peng[1], Bing Hu[3], Mieczyslaw Martin Marcinkiewicz[4], Alain Lamarre[2] & Jiangping Wu[1,5]

Armadillo repeat containing 5 (ARMC5) is a cytosolic protein with no enzymatic activities. Little is known about its function and mechanisms of action, except that gene mutations are associated with risks of primary macronodular adrenal gland hyperplasia. Here we map *Armc5* expression by *in situ* hybridization, and generate *Armc5* knockout mice, which are small in body size. *Armc5* knockout mice have compromised T-cell proliferation and differentiation into Th1 and Th17 cells, increased T-cell apoptosis, reduced severity of experimental autoimmune encephalitis, and defective immune responses to lymphocytic choriomeningitis virus infection. These mice also develop adrenal gland hyperplasia in old age. Yeast 2-hybrid assays identify 16 ARMC5-binding partners. Together these data indicate that ARMC5 is crucial in fetal development, T-cell function and adrenal gland growth homeostasis, and that the functions of ARMC5 probably depend on interaction with multiple signalling pathways.

[1] Centre de recherche (CR), Centre hospitalier de l'Université de Montréal (CHUM), 900 Rue Saint Denis, Montréal, Québec, Canada H2X 0A9. [2] Institut national de la recherche scientifique—Institut Armand-Frappier (INRS-IAF), 531 Boul. des Prairies, Laval, Québec, Canada H7V 1B7. [3] Anatomic Pathology, AmeriPath Central Florida, 4225 Fowler Ave. Tampa, Orlando, Florida 33617, USA. [4] Cytochem Inc., 6465 Av Durocher, Outremont, Montréal, Québec, Canada H2V 3Z1. [5] Nephrology Service, Centre hospitalier de l'Université de Montréal (CHUM), 900 Rue Saint Denis, Montréal, Québec, Canada H2X 0A9. * These authors contributed equally to this work. Correspondence and requests for materials should be addressed to J.W. (email: jianping.wu@umontreal.ca).

The gene Armadillo was first identified in the fruit fly *Drosophila melanogaster* as a gene controlling larval segmentation with morphological similarity to armadillos[1,2]. β-Catenin is the human and mouse orthologue of fruit fly Armadillo[3]. Armadillo/β-catenin protein contains 13 and 12 conserved armadillo (ARM) repeats, respectively: each repeat is about 40 amino acid (aa) long and consists of 3 α-helices[4]. Multiple repeats form an ARM domain which has a groove for binding various other proteins in its tertiary structure[5]. More than 240 proteins, from yeasts to humans, are known to contain an ARM domain[6,7]. Although β-catenin is believed to interact with and regulate cytoskeleton function, its roles and those of ARM domain-containing proteins, in general, are very versatile in cell biology, including cytoskeleton organization[8], cell-cell interactions[9], protein nuclear import[10], degradation[11] and folding[12], cell signalling/sensing[13–15], molecular chaperoning[16], cell invasion/mobility/migration[17], transcription control[18], cell division/proliferation[19] and spindle formation[20], to name some of them.

At the tissue and organ levels, ARM domain-containing proteins are involved in T-cell development[21], lung morphogenesis[22], limb dorsal-ventral axis formation[23], neural tube development[24], osteoblast/chondrocyte switch[25], synovial joint formation[26], adrenal gland cortex development[27] and tumour suppression[28].

Due to the very diverse functions of ARM domain-containing proteins, it is challenging to predict their mechanisms of action. Indeed, these aspects of many ARM domain-containing proteins remain undeciphered, and quite a number of them are given the name ARMC (ARM repeat-containing), followed by Arabic numbers (for example, ARMC1, 2, 3 and so on). ARMC5 is one such protein.

Human and mouse ARMC5 proteins share 90% aa sequence homology and have similar structures[29,30]. Mouse ARMC5 is 926 aa in length and contains 7 ARM repeats. A BTB/POZ domain towards its C-terminus is responsible for dimerization or trimerization[31–33]. Several groups reported in 2013 and 2014 that *ARMC5* gene mutations are associated with primary macronodular adrenal hyperplasia (PMAH) and Cushing's syndrome[34–36]. Assié *et al.*[34] demonstrated that no viable HeLa cells could be obtained when they were stably transfected with *ARMC5*-expressing vectors. They suggested that the default function of wild-type (WT) ARMC5 is the suppression of cell proliferation or promotion of apoptosis, which might explain the adrenal cortex hyperplasia seen in patients with ARMC mutations. No other reports on ARMC5 function and mechanisms of action are available in the literature.

In the present work, we have studied the tissue-specific expression of *Armc5*. We have generated *Armc5* gene knockout (KO) mice, and revealed that ARMC5 is vital in development and immune responses. We also show that aged KO mice develop adrenal gland hyperplasia. We have identified a group of ARMC5-interacting proteins by yeast 2-hybrid (Y2H) assay, paving the way for further mechanistic and functional investigations of ARMC5.

## Results

**Armc5 expression in mice and T cells.** *Armc5* mRNA expression was analysed by *in situ* hybridization (ISH) in adult WT mice. Haematoxylin/eosin staining of a consecutive sagittal whole body section preceded ISH (Fig. 1a, upper panel). *Armc5* expression, based on anti-sense riboprobe hybridization (Fig. 1a, middle panel), was high in the thymus, stomach, bone marrow and lymphatic tissues (including lymph nodes and intestinal wall). The hybridization was also apparent in the adrenal gland and skin. Some hybridization occurred in brain structures, with noticeable levels found in the cerebellum. Control hybridization with sense (S) riboprobes revealed a faint nonspecific background (Fig. 1a, bottom panel).

At the anatomical level, *Armc5* expression was high in the thymus cortex (Fig. 1b, upper left panel). This was confirmed at the microscopic level (Fig. 1b, bottom left panel). The higher *Armc5* signals in the cortex than in the medulla were due to higher cell density in the former. Sense riboprobes detected little background noise (Fig. 1b, right column). Based on reverse transcription-quantitative polymerase chain reaction (RT-qPCR) results, thymic stroma cells (including epithelial cells) had *Armc5* expression similar to that of thymocytes (Supplementary Fig. 1a). Further, there was no significant difference in *Armc5* expression among thymocyte subpopulations (CD4/CD8 double-negative (DN) 1–4, CD4/CD8 double-positive (DP), and CD4- or CD8-single-positive (SP); Supplementary Fig. 1b,c; gating strategy: Supplementary Fig. 2a), and between naive and memory spleen T cells (Supplementary Fig. 1d; gating strategy: Supplementary Fig. 2b).

Moderately intense *Armc5* labelling was apparent in spleen white pulp but not in red pulp (Fig. 1c, upper panel). At the microscopic level, small groups of cells in WP displayed *Armc5* signals (Fig. 1c bottom panel). Sense riboprobes detected no signals (Fig. 1c middle panel).

*Armc5* mRNA expression was induced rapidly in CD4 cells in 2 h after CD3ε plus CD28 stimulation, then subsided and remained low between 24 and 72 h post-activation (Fig. 1d, upper row). CD8 T cells had less *Armc5* induction and the levels remained low between 24 and 72 h (Fig. 1d, lower row).

ARMC5 was mainly a cytosolic protein, as it was detected in the cytoplasm of L cells transiently transfected with a mouse ARMC5 expression construct (Fig. 1e,f).

**Generation of Armc5 KO mice.** We generated *Armc5* KO mice to understand the biological roles of ARMC5 in general and T-cell-mediated immune responses in particular. Our targeting strategy is illustrated (Fig. 2a). Germline transmission was confirmed by Southern blotting of tail DNA (Supplementary Fig. 3). With the 5′ end probe, the WT allele after EcoRV digestion gave a 9.3-kb band, and the KO allele, a 6.6-kb band (Supplementary Fig. 3, upper panel). With the 3′ end probe, the WT allele after HindIII digestion presented a 12.5-kb band, and the KO allele, an 8.7-kb band (Supplementary Fig. 3, lower panel). WT (mice 3 and 7) and heterozygous mice (mice 1, 2, 4, 5 and 6) were thus identified. Mouse 1 in the original 129/sv × C57BL/6J background was backcrossed to different genetic backgrounds for experimentation, as detailed below.

*Armc5* deletion of KO mice at the mRNA level in spleen T cells, thymocytes, lymph nodes, brain and adrenal glands was confirmed by RT-qPCR (Fig. 2b).

**General phenotype of Armc5 KO mice.** When *Armc5* KO mice were in the C57BL/6J × 129/sv F1 background, only about 10% live KO pups were delivered in a heterozygous × heterozygous mating strategy, below the expected 25% Mendelian rate. After F1 mice were backcrossed to C57BL/6 for five or more generations, no KO pups could be produced, nor were live KO pups born after the mice were backcrossed eight generations to the 129/sv background. This suggested that *Armc5* deletion caused embryonic lethality, with its severity depending on genetic background of the mice: embryonic lethality became more severe with higher degrees of genetic background purity. KO mice in the C57BL/6J × 129/sv F1 background were studied in subsequent experiments.

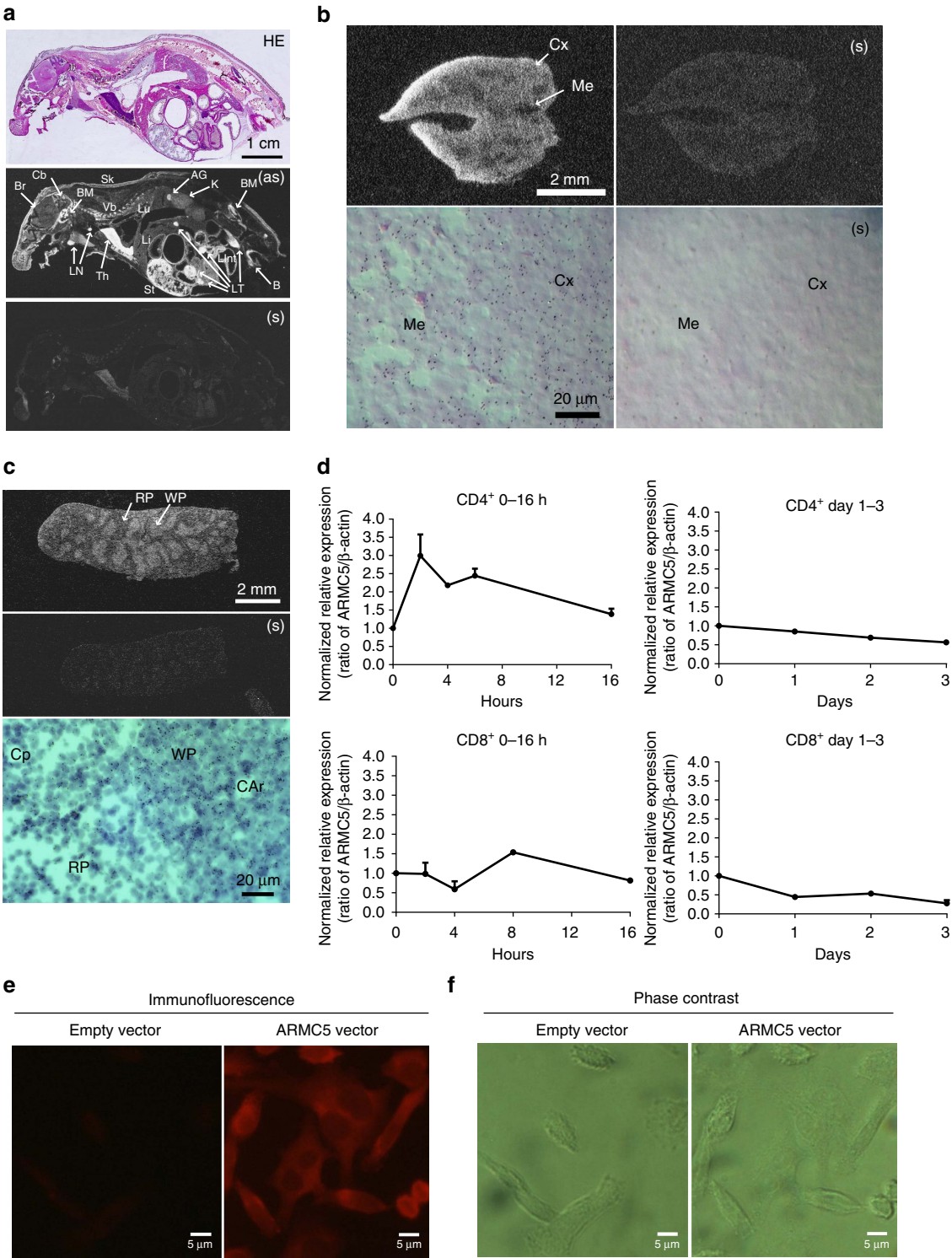

**Figure 1 | Armc5 tissue-specific expression: *Armc5* mRNA expression in mice is assessed by ISH. (a)** *Armc5* expression in adult mouse using whole-body sections. Upper panel: H/E staining; middle and bottom panels: dark field X-ray film autography with anti-sense (AS) cRNA or sense (S) cRNA as probes, respectively. Bar = 1 cm. AG: adrenal gland; B: bone; BM: bone marrow; Cb: cerebellum; K: kidney; Lint: large intestine; LT: lymphatic tissue; Sk: skin; ST: stomach; Th: thymus; VB: vertebrae. **(b)** *Armc5* expression in the adult thymus. Upper row: dark field X-ray film autography; lower row: bright field emulsion autoradiography; left column: anti-sense probe; right column: sense probe. Bars = 2 mm and 20 μm. Cx: cortex; Me: medulla. **(c)** *Armc5* expression in the adult spleen. Upper and middle panels: dark field X-ray film autography, with anti-sense and sense probes, respectively; bottom panel: bright field emulsion autoradiography. Bars = 2 mm and 20 μm. WP: white pulp; RP: red pulp; CAr: central artery; Cp: capillary. **(d)** *Armc5* mRNA in mouse spleen CD4 and CD8 cells measured by RT-qPCR. Experiments were performed three times. The results of representative experiments are shown. To facilitate comparison, normalized ratios of *Armc5* versus β-actin signals (means ± s.e.m.) are presented; the 0-h signal ratio of each experiment is considered as 1. **(e)** ARMC5 subcellular localization in L cells was detected immunofluorescence. L cells were transfected with HA-tagged mouse ARMC5-expressing construct or an empty vector, as indicated. **(f)** Phase contract micrographs of views in **(e)**. The experiments were conducted three times, and micrographs of a representative experiment are shown. Scale bar: 5 μm.

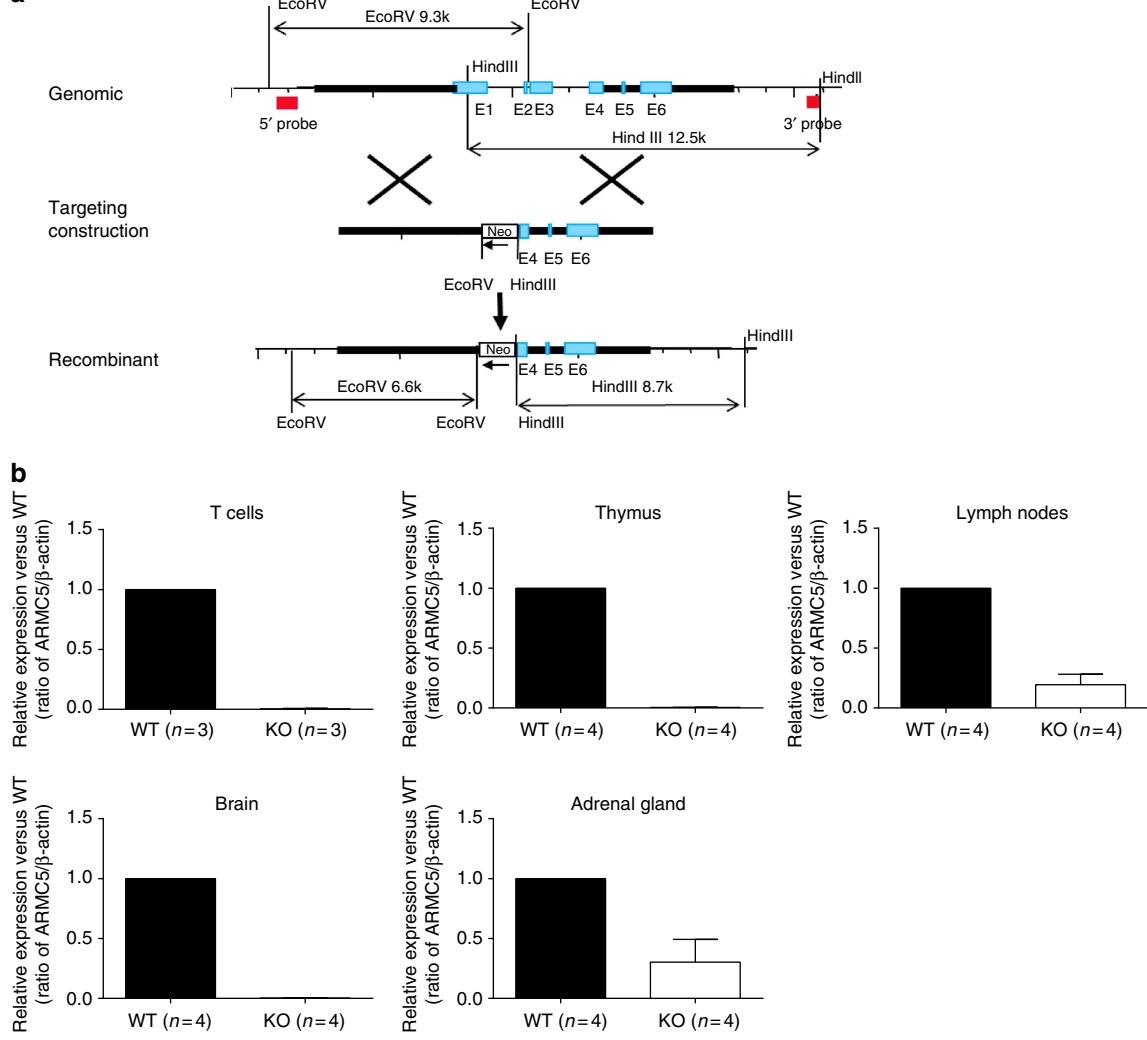

**Figure 2 | Generation of Armc5 KO mice.** (**a**) *Armc5* KO mice were generated by targeted gene deletion. The targeting strategy is depicted. Red squares on 5′ and 3′ sides of the mouse *Armc5* WT genomic sequence represent sequences serving as probes for genotyping by Southern blotting. (**b**) *Armc5* mRNA deletion in KO mice was confirmed by RT-qPCR. The results are expressed as normalized ratios (means ± s.e.m.) of *Armc5* versus *β-actin* mRNA signals. The values from WT mice are considered as 1. Experiments were conducted more than three times, and representative results are reported.

KO embryos were smaller than WT controls at embryonic day 14 (Fig. 3a). These KO pups were smaller at age 8–12 weeks (Fig. 3b). Body weight was significantly lower in KO and WT mice at age 4 and 8 weeks than in their WT littermates (Fig. 3c). Both male and female KO mice weighed only about 60% as much as WT controls.

We examined serum growth hormone levels because of growth retardation in KO mice, but no significant difference was found between them and their WT counterparts (Supplementary Fig. 4).

*ARMC5* gene mutations have been reported to be linked with PMAH and Cushing's syndrome[34]. However, KO mice presented normal adrenal gland size and histology (Supplementary Fig. 5) and serum glucocorticoid levels (Supplementary Fig. 6) in young age (less than age 5 months). In old age (>15 months), grossly, KO mice showed enlarged adrenal glands without apparent nodular structure, and histologically, there is no identifiable nodular hyperplasia (Fig. 3d). Serum glucocorticoid levels were significantly increased in aged KO mice (Fig. 3e), supporting the notion that the adrenal gland hyperplasia is of cortex in nature. It is to be noted that the mice were killed between 12:30 and 13:30, and their blood was harvested for the measurement of glucocorticoids, whose secretion is at the nadir at this time point.

The moderate but significant increase of glucocorticoid levels in the KO mice is reminiscent of human PMAH, in which the increase of glucocorticoid levels is not drastic and is caused by the large mass of the adrenal gland, while on a per cell basis, the secretion is reduced[34].

**Armc5 KO phenotype in lymphoid organs and T cells.** Thymus (Supplementary Fig. 7a) and spleen (Supplementary Fig. 7b) weight and cellularity were not significantly different in KO and WT mice. Moreover, thymocyte sub-populations (CD4+CD8+ DP, CD4+ SP and CD8+ SP cells) in the KO and WT thymus were comparable (Supplementary Fig. 7c), as were spleen lymphocyte subpopulations (Thy1.2+ T cells versus B220+ B cells; CD4+ versus CD8+ T cells; Supplementary Fig. 7d).

Despite seemingly normal T-cell development in KO mice, T-cell proliferation triggered by anti-CD3ε was compromised in both CD4 and CD8 cells (Fig. 4a, left and middle panels; gating strategy: Supplementary Fig. 2c). It is to be noted that activation markers CD25 and CD69 shortly after CD3 stimulation were drastically upregulated and were always comparable between WT and KO T cells (Supplementary Fig. 8). The proliferation rate of

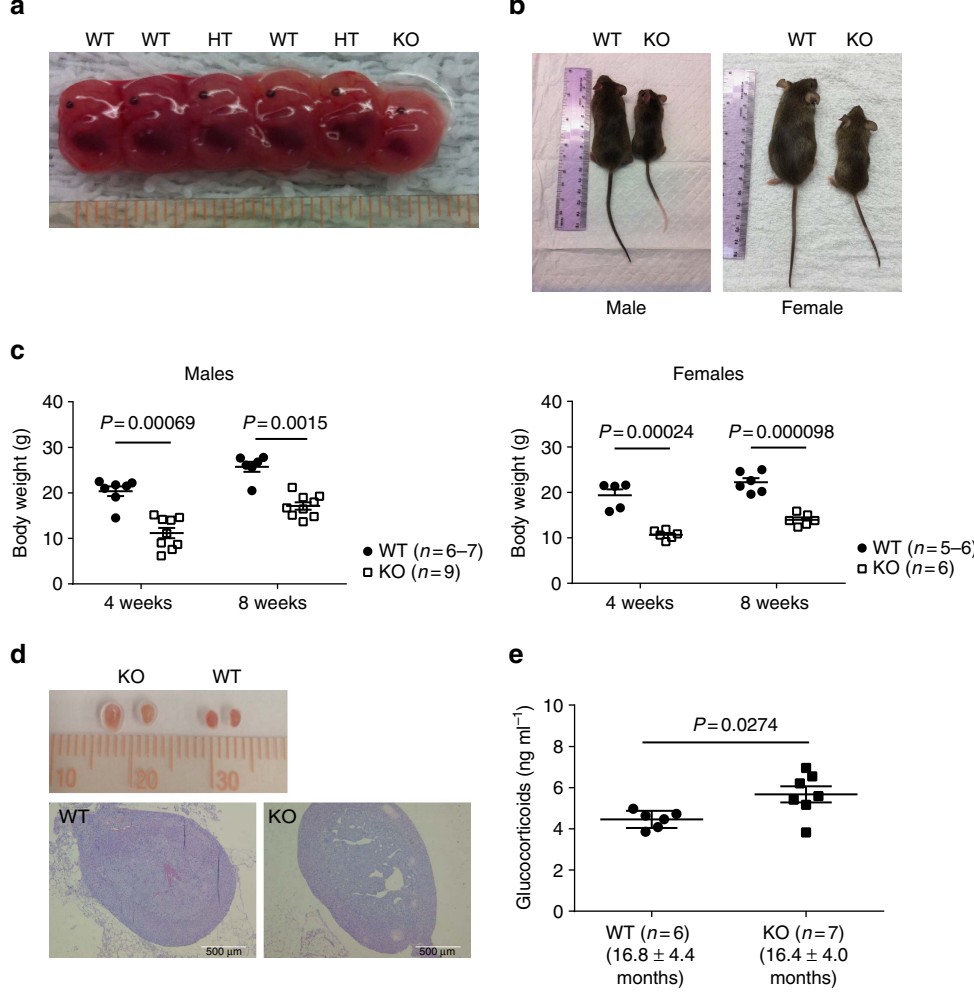

**Figure 3 | General phenotype of KO mice. (a)** Representative photos of WT, HT, KO fetuses on embryonic day 14. **(b)** Representative photos of adult KO and WT littermates. Left panel: males (8 weeks old); right panel: females (12 weeks old). **(c)** Body weight (means ± s.e.m.) of *Armc5* KO and WT littermates at age 4 and 8 weeks. Mouse numbers (*n*) per group are indicated. *$p < 0.001$ (two-tailed Student's *t*-test). **(d)** Morphology (upper panel) and histology (lower panel, HE staining) of adrenal glands from old KO mice (19 months old). Magnification: × 5. Scale bar: 500 μm. **(e)** Serum glucocorticoid levels in old KO mice. Means ± s.e.m. of serum glucocorticoids in old KO and WT mice are shown. Age of each group (means ± s.e.m.) and mouse number per group are indicated. Two-tailed Student's *t*-test was used for statistical analysis.

KO B cells was also lower (Fig. 4a, right panel; gating strategy: Supplementary Fig. 2d). Cell cycle analysis revealed that G1/S progression was compromised in KO T cells (Fig. 4b; gating strategy: Supplementary Fig. 2e). We also demonstrated that KO T cells (gated on $CD4^+$ plus $CD8^+$ cells) presented increased FasL-triggered apoptosis (Fig. 4c; gating strategy: Supplementary Fig. 2f).

Naive KO CD4 cells cultured under Th1 and Th17 conditions manifested reduced proliferation, as expected (Fig. 5a; gating strategy: Supplementary Fig. 2g). The differentiation of naive CD4 cells into Th1 and Th17 cells was defective (Fig. 5b; gating strategy: Supplementary Fig. 2g), since the percentages of Th1 or Th17 cells were decreased among CD4 cells, which had already proliferated. The expression of transcription factors T-bet and RORγt—essential for Th1 and Th17 differentiation, respectively—were normal in KO CD4 cells cultured under Th1 and Th17 differentiation conditions (Fig. 5c,d), when gated on either total CD4 cells or on those already differentiated cells ($IFN-\gamma^+$ or $IL-17^+$ cells), suggesting that the defective differentiation is not caused by a lack of these transcription factors.

As for humoral immune responses, KO serum IgG levels were comparable to those of WT controls (Supplementary Fig. 9).

We generated chimeric mice by transplanting KO and WT fetal liver cells in the C57BL/6J × 129/sv F1 background ($CD45.2^+$ SP) into lethally irradiated C57BL/6J × C57B6.SJL F1 mice ($CD45.1^+ CD45.2^+$ DP). Peripheral white blood cells of the recipients were analysed by flow cytometry 8 weeks after transplantation, and recipients of similar degrees of KO and WT chimerism were paired for experimentation. Typically, about 80–85% of peripheral white blood cells were of donor origin ($CD45.2^+$ SP), and 12–15%, of recipient origin ($CD45.1^+ CD45.2^+$ DP). In spleen $Thy1.2^+$ total T cells, $CD4^+$ T cells and $CD8^+$ T cells, 60–70% were of donor origin, and 30–35% of recipient origin (Supplementary Fig. 10). Unlike in *Armc5* KO mice, KO T cells in chimeras were developed in a WT environment, devoid of influence by putatively unknown factors which might exist in the total KO environment and have aberrant effects on T-cell development.

We showed that donor-derived KO naive CD4 cells were defective in differentiating into Th1 cells (Fig. 5e; gating strategy: Supplementary Fig. 2h), similar to CD4 cells from unmanipulated, naive KO mice (Fig. 5b). The KO Th17 cell differentiation in this model was also compromised, although did not reach statistical significance, probably due to an inadequate sample size (Fig. 5e).

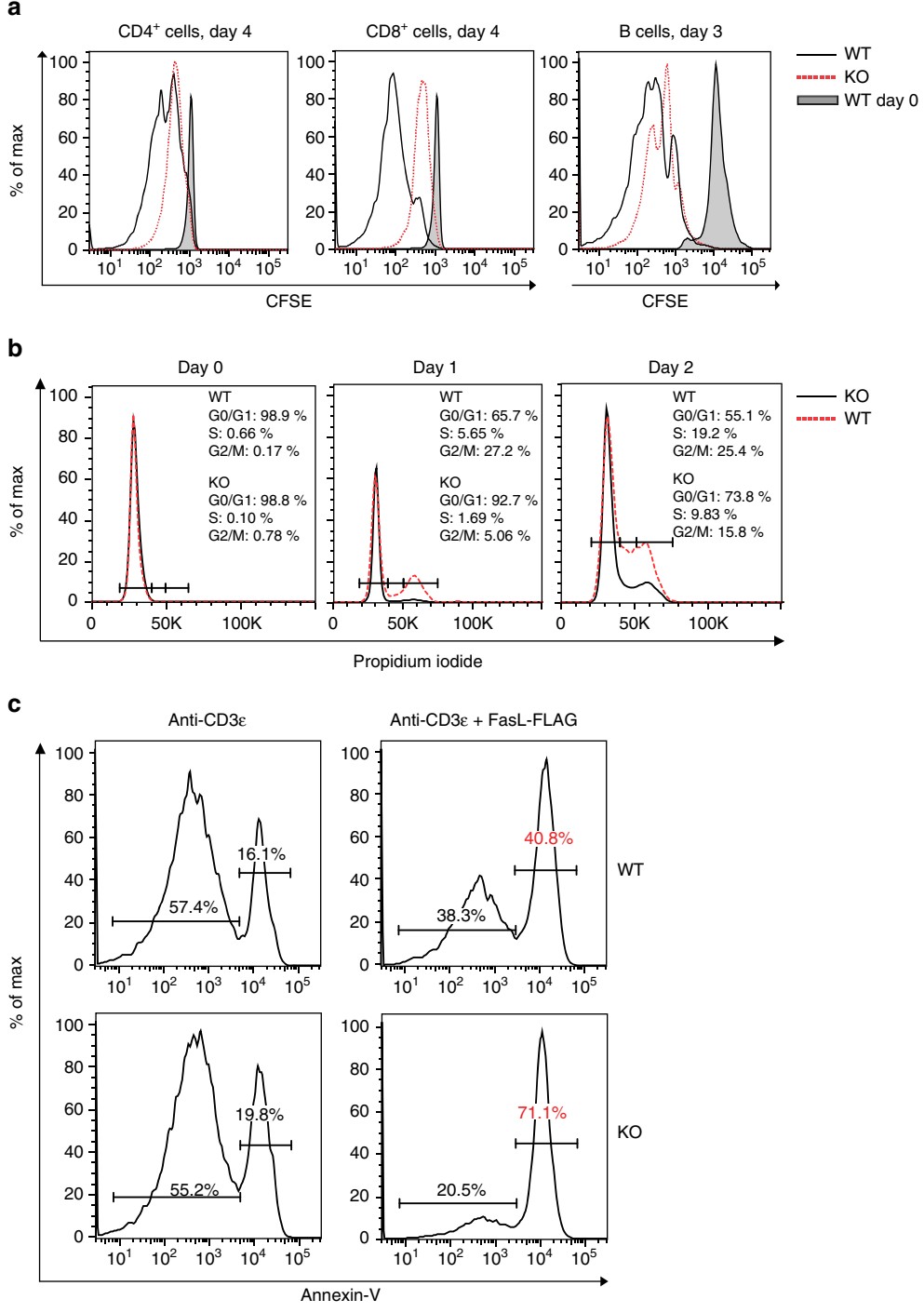

**Figure 4 | KO T-cell proliferation and apoptosis.** (**a**) Proliferation of spleen CD4$^+$ and CD8$^+$ T cells and B220$^+$ B cells from WT and KO mice according to CFSE staining. CFSE intensity was ascertained by flow cytometry. Experiments were conducted independently 4–6 times. Representative histograms are shown. (**b**) Cell cycle progression of spleen T cells from WT and KO mice. The percentages of cells in G$_1$, S and G$_2$ phases are indicated. Experiments were conducted independently three times. Representative histograms are shown. (**c**) Apoptosis of WT and KO spleen T cells (gated on CD4$^+$ plus CD8$^+$ cells) upon FasL stimulation was determined by their annexin V expression according to flow cytometry. Experiments were conducted independently three times. Representative histograms are shown.

**Experimental autoimmune encephalomyelitis (EAE) in KO mice.** To understand the role of ARMC5 in *in vivo* T-cell immune responses, particularly CD4-mediated immune responses, we induced experimental autoimmune encephalomyelitis (EAE) in *Armc5* WT and KO mice. As shown in Fig. 6a, WT mice started to manifest clinical signs of EAE on day $13.2 \pm 1.30$ (means ± s.e.m.) after immunization, and their symptoms peaked

on day 23. The onset of clinical symptoms in KO mice was delayed by about 7 days, and their maximum disease score was significantly lower than that of WT controls ($P < 0.01$, two-tailed Student's *t*-test) after day 18. Disease incidence was lower in KO mice between days 15 and 18, although it reached 100% in both KO and WT groups after day 28 (Fig. 6b). A trend towards less body weight loss in KO mice was noted after EAE induction

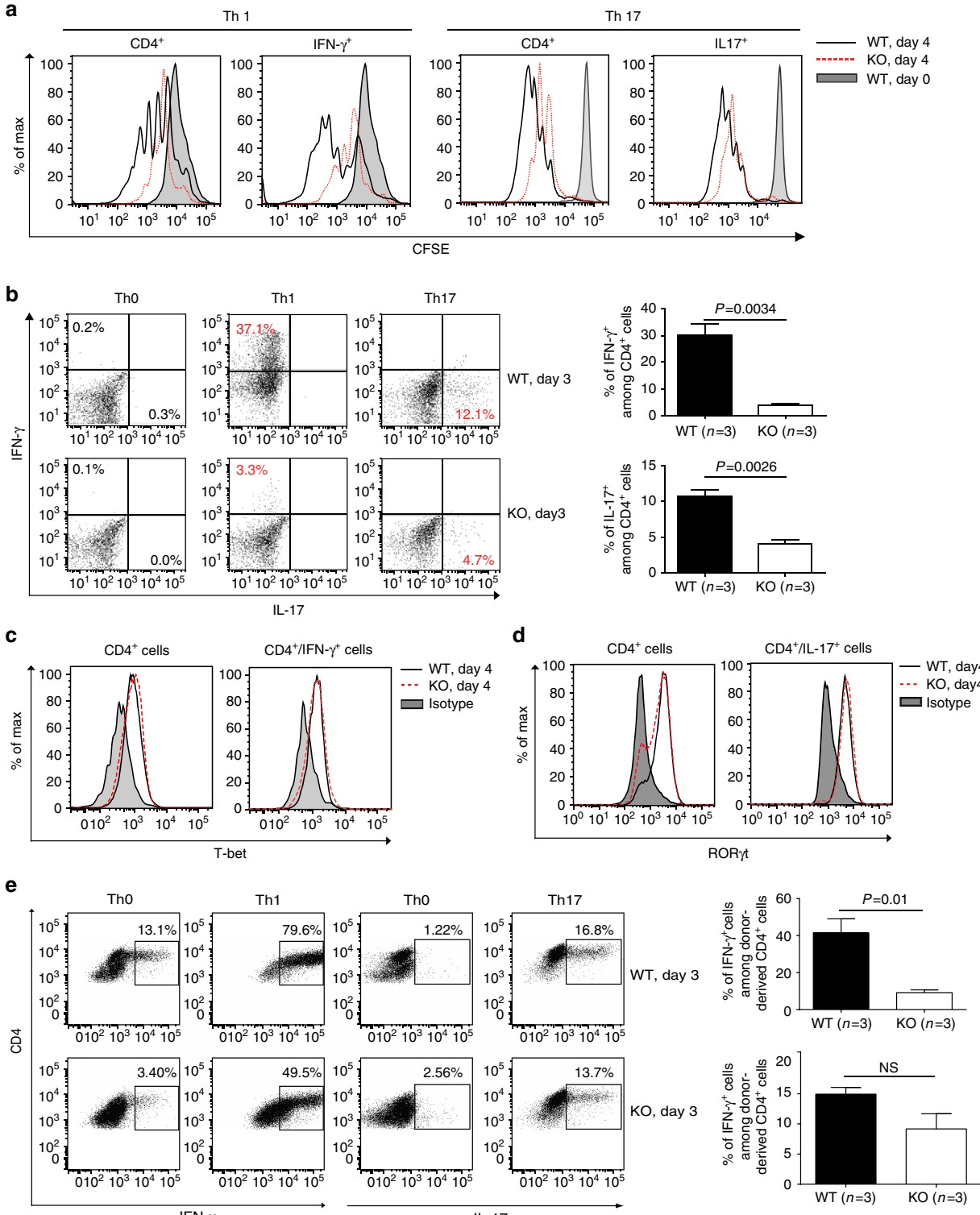

**Figure 5 | Proliferation and differentiation of naive KO CD4 cells into Th1 and Th17 cells.** (**a**) Proliferation of WT and KO naive spleen CD4 cells under Th1 and Th17 conditions was assessed based on CFSE content according to flow cytometry. Experiments were conducted three times, and representative histograms are shown. Grey peaks represent the CFSE content of CD4 cells at day 0. (**b**) These cells' differentiation into Th1 and Th17 cells was also determined by flow cytometry according to intracellular IFN-γ and IL-17 positivity (gated on total CD4$^+$). Representative dot plots are shown in the left panel. Means ± s.e.m. of data from three experiments are presented as bar graphs in the right panel. Mouse numbers (*n*) per group are indicated. *P* values are reported in the bar graphs (two-tailed Student's *t*-test). (**c,d**) T-bet and RORγt expression in CD4 cells cultured under Th1 and Th17 conditions or in IFNγ$^+$ or IL-17$^+$ cells was determined by flow cytometry. Experiments were conducted three times. Representative histograms are shown. (**e**) Th1 and Th17 differentiation of naive spleen CD4 cells (CD45.2 single-positive) derived from WT and KO donors in chimeric mice was analysed by flow cytometry based on their intracellular IFN-γ and IL-17 expression. Representative dot plots are shown in the left panel. Means ± s.e.m. of data from three experiments are presented as bar graphs in the right panel. Mouse numbers (*n*) per group are indicated. *p* values are reported in the bar graphs (two-tailed Student's *t*-test).

compared to WT controls, but statistically significant difference was reached only on day 22 (Fig. 6c).

KO mice had significantly fewer cells in their draining lymph node (LN) and fewer infiltrating mononuclear cells in the brain and spinal cords on day 14 after myelin oligodendrocyte

glycoprotein (MOG) immunization compared to WT controls (Fig. 6d). After *ex vivo* phorbol 12-myristate 13-acetate (PMA)/ ionomycin stimulation, the percentage of interferon-gamma (IFN-γ⁺) CD4 cells among total CD4 cells from the LN of KO mice was significantly lower than that of WT mice (6.2 versus

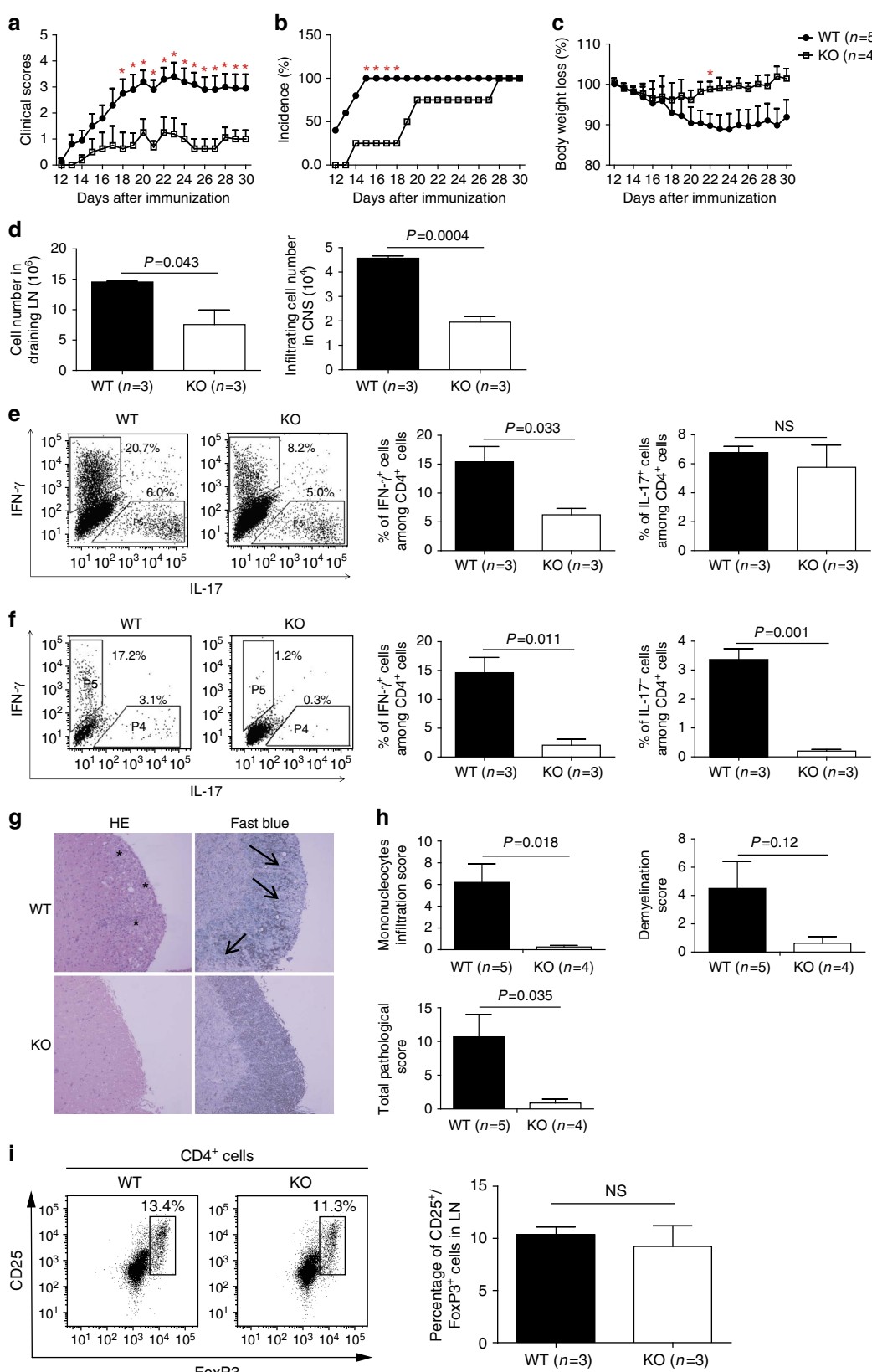

15.4%), although the percentage of IL-17$^+$ cells among CD4 cells was similar in KO and WT draining LN (Fig. 6e). The percentages of IFN-$\gamma^+$ and IL-17$^+$ populations in CD4$^+$ T cells from the central nervous system (CNS) of KO mice were significantly lower after *ex vivo* PMA/ionomycin stimulation than in WT mice (Fig. 6f).

Histologically, spinal cords from KO animals on day 30 after MOG immunization showed less severe mononuclear cell infiltration and demyelination, according to HE and Luxol Fast Blue staining, respectively, compared to their WT counterparts (Fig. 6g). Histological data from four KO and five WT spinal cords are summarized (Fig. 6h). Mononuclear cell infiltration in KO spinal cords was significantly lower than in WT controls. Although demyelination in the former was also lower, it did not reach statistical significance. However, combined pathological scores, which included degrees of both mononuclear cell infiltration and demyelination, were significantly lower in KO mice. We did not observe changes in the percentages of Treg cells in the spleen of naive KO mice or in the draining LN of KO mice on day 17 during EAE induction, compared to WT controls (Fig. 6i; gating strategy: Supplementary Fig. 2i). Therefore, it is unlikely that Treg cells are implicated in reduced EAE severity in KO mice.

To exclude the possible influence of the *Armc5* KO background on the immune system of KO mice, EAE was also induced in chimeras transplanted with fetal liver cells from *Armc5* WT and KO embryos on days 13–15. Overall, KO chimeras still displayed a lower degree of EAE than WT chimeras, but the difference was not as dramatic as in real KO versus WT mice. The onset of clinical symptoms in KO chimeras occurred 2.5 days (mean) later than in WT chimeras. KO chimera clinical scores tended to be lower than those of WT mice, but were only significantly different between days 12 and 14 (Fig. 7a). EAE incidence was significantly lower on days 11, 12 and 14 after immunization (Fig. 7b). A trend of less body weight loss was noted in KO chimeric mice, although no statistical difference was apparent between the KO and WT groups (Fig. 7c). When stimulated *ex vivo* by PMA/ionomycin, the percentage of KO donor-derived IFN-$\gamma^+$ CD4 cells among total KO donor-derived CD4 cells from the CNS was significantly lower than in WT controls (Fig. 7d; gating strategy: Supplementary Fig. 2h), as was the case in real KO mice. However, there was no significant difference between the percentage of KO donor-derived IL-17$^+$ CD4 cells among total KO donor-derived CD4 cells and that of WT mice. The reduced degree of difference in EAE manifestation in KO versus WT chimeras, compared to that in real KO versus WT mice,

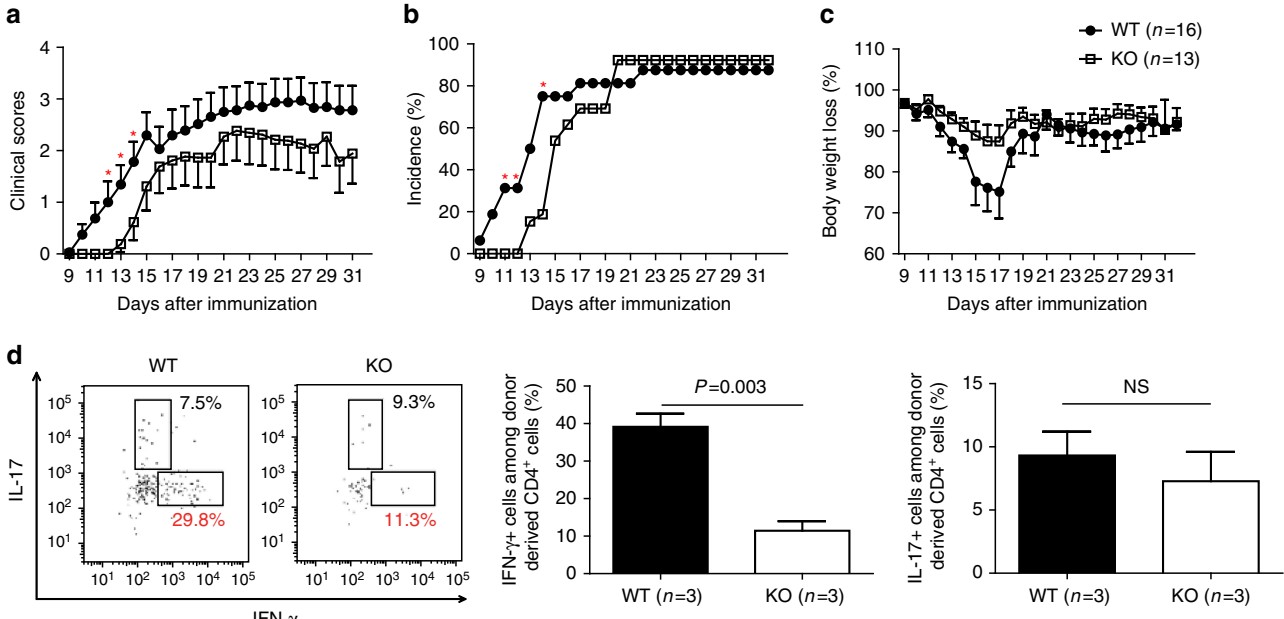

**Figure 7 | EAE induction in chimeric KO mice.** (**a**) Means ± s.e.m. of EAE clinical scores of chimeric mice.*P<0.05 (two-tailed Student's *t*-test). (**b**) EAE incidence in chimeric mice. *P<0.05 (chi-square test). (**c**) Means ± s.e.m. of body weight of chimeric mice with body weight on day 10 after MOG immunization considered as 100%. No significant difference is found (two-tailed Student's *t*-test). (**d**) Cytokine-producing donor-derived CD4 cells in the CNS of chimeric mice on day 14 after MOG immunization. Left panel: representative dot plots; right panel: summary (means ± s.e.m.) of all the results, with mouse numbers (*n*) and *P*-values (paired two-tailed Student's *t*-test) indicated.

**Figure 6 | EAE induction in KO mice.** (**a**) Means ± s.e.m. of EAE clinical scores of KO and WT mice. *P<0.05 (two-tailed Student's *t*-test). (**b**) EAE incidence in KO and WT mice. *P<0.05 (chi-square test). (**c**) Means ± s.e.m. of body weight of KO and WT mice during EAE induction. Body weight of mice on day 10 post-immunization was considered as 100%. *P<0.05 (two-tailed Student's *t*-test). (**d**) Means ± s.e.m. of cellularity in draining LN and of cells infiltrating the CNS of mice 14 days after MOG immunization. Mouse numbers (*n*) and *P* values (paired two-tailed Student's *t*-test) are indicated. (**e**,**f**) Cytokine-producing cells among CD4 cells from draining LN (**e**) and CNS (**f**) on days 13–18 after MOG immunization. Left panels: representative dot plots; right panel: bar graphs (means ± s.e.m.) summarizing all the results, with mouse numbers and *P* values (two-tailed Student's *t*-test) indicated. (**g**) HE (left column) or Luxol Fast Blue (right column) staining of spinal cords 30 days after MOG immunization. Asterisks indicate cell infiltration. Arrows point to demyelination. (**h**) Means ± s.e.m. of mononuclear cell infiltration scores, demyelination scores and total pathological scores, which is the sum of the first two scores. Mouse numbers (*n*) and *P* values (two-tailed Student's *t*-test) are indicated. (**i**) Treg cells in naive KO mice on day 17 during EAE induction. Left panel: representative dot plots; right panel: means ± s.e.m. of data from three experiments. NS: not significant (two-tailed Student's *t*-test).

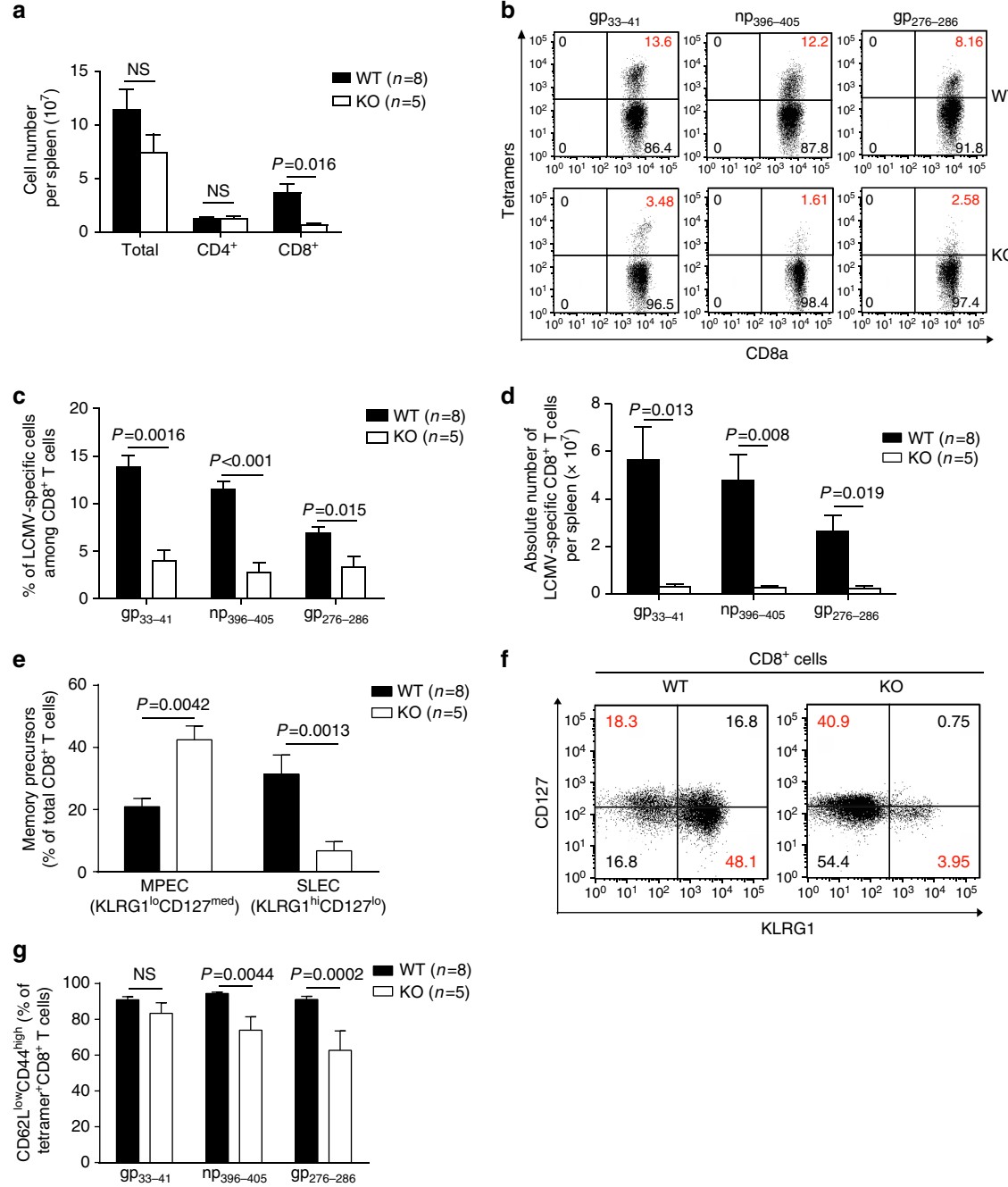

**Figure 8 | T-cell expansion in KO mice after LCMV infection.** (**a**) Spleen CD8 cell numbers in KO mice on day 8 after LCMV infection as determined by flow cytometry. Mice number (*n*), means ± s.e.m. and *P* values (two-tailed Student's *t*-test) are indicated. (**b**) Virus-specific spleen CD8 cells in KO mice on day 8 post-LCMV infection according to flow cytometry. Representative dot plots are shown. (**c**) Means ± s.e.m. of percentages of gp33–41, np396–405 and gp276–286 tetramer-positive cells among spleen CD8 cells from all the results are presented. Numbers (*n*) of mice per group and *P* values (two-tailed Student's *t*-test) are indicated. (**d**) Means ± s.e.m. of absolute numbers of gp33–41, np396–405 and gp276–286 tetramer-positive CD8 cells in the KO and WT mouse spleens on day 8 post infection. Numbers (*n*) of mice per group and *P* values (two-tailed Student's *t*-test) are indicated. (**e,f**) Memory and effector CD8 cell maturation in LCMV-infected WT and KO mice on day 8 post LCMV infection. KLRG1loCD127hi cells are considered as memory precursor effector cells (MPEC), and KLRG1hiCD127lo cells as short-lived effector cells (SLEC). Means ± s.e.m. are presented. Numbers (*n*) of mice per group and *P* values (two-tailed Student's *t*-test) are indicated (**e**). Representative dot plots are shown (**f**). (**g**) On day 8 post infection, total gp33–41 np396–405 and gp276–286 tetramer-positive CD8 cells in KO and WT mouse spleen were assessed for activation markers. Means ± s.e.m. are presented. Numbers (*n*) of mice per group and *P* values (two-tailed Student's *t*-test) are indicated.

was not unexpected, as KO chimeras contained about 30% recipient-derived T cells, which were fully immuno-competent WT T cells.

**Antiviral immune responses in KO mice.** CD8 T-cell-mediated immune responses play a critical role against lymphocytic

choriomeningitis virus (LCMV) infection. Therefore, we assessed KO CD8$^+$ T-cell functions in LCMV infection. Eight days after mice were infected with LCMV (strain WE), absolute numbers of WT CD8$^+$ T cells but not CD4$^+$ T cells increased significantly (Fig. 8a; gating strategy: Supplementary Fig. 2j). LCMV tetramer staining showed that both the number of gp$_{33-41}$ –, np$_{396-405}$ –

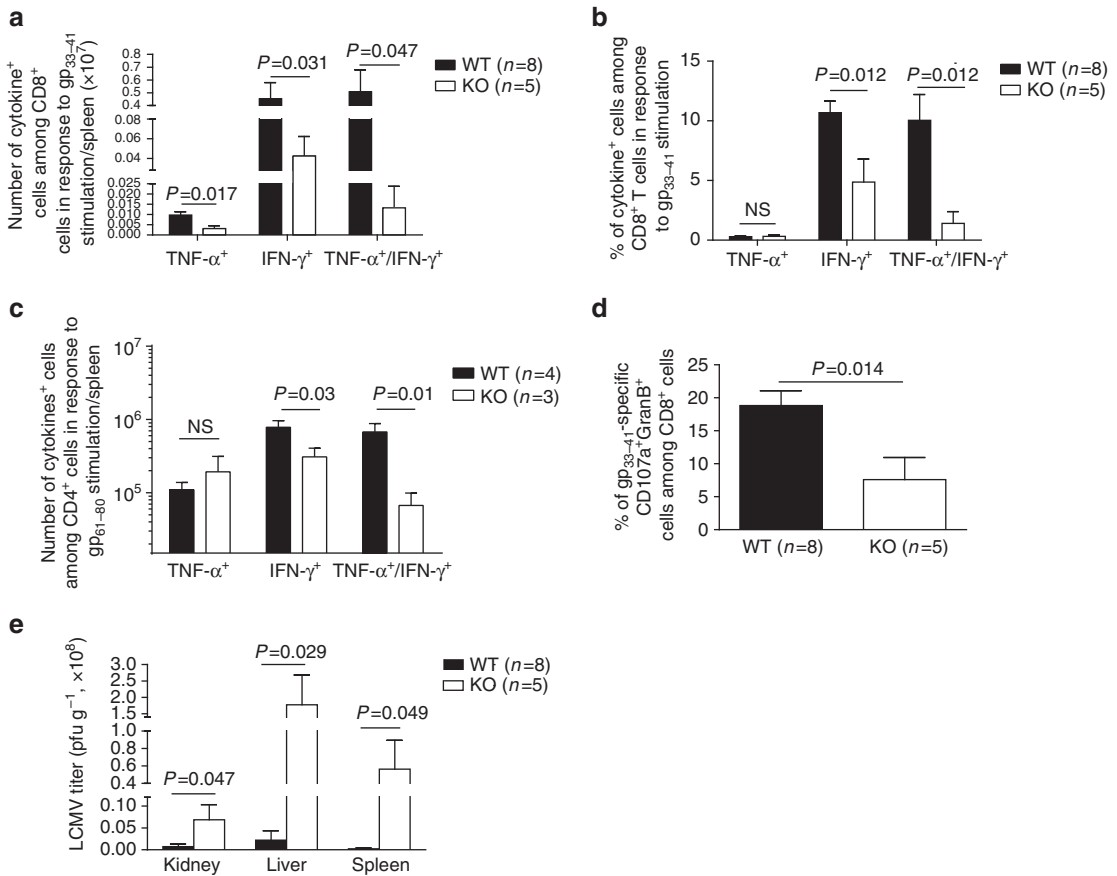

**Figure 9 | T-cell cytokine production and virus titres in KO mice after LCMV infection. (a)** Absolute number of virus-specific, cytokine-producing CD8 cells. **(b,c)** Percentages of virus-specific, cytokine-producing cells among CD8 cells **(b)** and CD4 cells **(c)** on day 8 post LCMV infection. Means ± s.e.m. of data are shown. Mouse numbers ($n$) per group and $P$ values (two-tailed Student's $t$-test) are indicated. **(d)** Means ± s.e.m. of percentages of $gp_{33-41}$-specific CD107a+GranB+ CD8 T cells on day 8 post LCMV infection. Mouse numbers ($n$) per group and $P$ values (two-tailed Student's $t$-test) are indicated. **(e)** Means ± s.e.m. of viral titres in the kidney, liver and spleen on day 8 post LCMV infection. Mouse numbers ($n$) per group and $P$ values (two-tailed Student's $t$-test) are indicated.

and $gp_{276-286}$-specific CD8 T cells per spleen and their percentage among total spleen CD8+ T cells were significantly lower in KO than in WT mice (Fig. 8b–d; gating strategy: Supplementary Fig. 2k), suggesting compromised CD8 cell clonal expansion after viral Ag stimulation in KO mice.

After infection, CD8 cells develop into KLRG1hiCD127lo short-lived effector cells (SLEC) and KLRG1loCD127med memory precursor effector cells (MPEC)[35]. In KO mice, 8 days after LCMV infection, the percentage of SLEC among CD8 T cells was significantly lower (Fig. 8e,f; gating strategy: Supplementary Fig. 2k), indicating defective anti-virus effector cell development. At the same time, MPEC percentage among CD8 T cells was increased in KO mice. The significance of this finding is not clear at present, although the percentage of CD62LloCD44hi effector memory cells among total CD8 cells (Supplementary Fig. 11a) and LCMV subdominant epitope ($np_{396-405}$ and $gp_{276-286}$)-specific CD8 T cells (Fig. 8g; gating strategy: Supplementary Fig. 2k) in KO mice was reduced.

We next examined the presence of LCMV-specific, cytokine-producing splenic T cells in virus-infected mice. As seen in Fig. 9a (gating strategy: Supplementary Fig. 2k), the absolute number of $gp_{33-41}$-specific TNF-α-positive CD8+ T cells, IFN-γ-positive CD8+ T cells and IFN-γ/TNF-α DP CD8 T cells per spleen was significantly lower in KO than in WT mice 8 days post-infection. Significantly lower percentages of $gp_{33-41}$-specific, IFN-γ-positive CD8 T cells and IFN-γ/TNF-α DP CD8+ T cells, but not TNF-α-

positive CD8 T cells, among total spleen CD8 T cells, were found in KO spleens (Fig. 9b, representative dot plots shown in Supplementary Fig. 11c). Similarly decreased numbers and percentages of LCMV-specific cytokine-producing cells were observed in the CD4 cell population, although the reduction was of lower magnitude compared to those in the CD8 cells (Fig. 9c; representative dot plots shown in Supplementary Fig. 12d).

In addition, lower percentages of $gp_{33-41}$-specific CD107a+ GranB+ T cells among total CD8+ T cells were observed in KO spleen (Fig. 9d; representative dot plots shown in Supplementary Fig. 11f), implying the presence of fewer functional virus-specific cytotoxic CD8+ T cells in KO mice. Virus titers in the kidneys, liver and spleen were significantly higher in KO mice 8 days post-LCMV infection, suggesting compromised virus clearance (Fig. 9e).

**Identification of ARMC5-binding proteins by Y2H assay.** ARMC5 has no enzymatic activity: its functions depend on interaction with molecules involved in different signalling pathways. To identify ARMC5-binding proteins, we conducted Y2H assays with human ARMC5 protein (Glu30-Ala935) as bait, and a human primary thymocyte cDNA expression library as prey. The binding proteins were given Predicted Biological Confidence scores[36], and 16 proteins with scores between A and D ('A' having the highest confidence of binding) are found (Table 1), if their coding sequences are in-frame and have no

**Table 1 | ARMC5-binding proteins identified by Y2H assay.**

| Gene name | PBC score | Binding clones | Different clones | Major known function |
|---|---|---|---|---|
| DAPK1 | A | 13 | 4 | Tumour suppressor, apoptosis, autophagy |
| ARMC5 | B | 2 | 2 | Self-dimerization |
| STK24 | B | 3 | 3 | Apoptosis, upstream of MAPK, acts on Tao |
| TTF1 | B | 3 | 2 | Transcription terminator, apoptosis, tumour risk |
| POLR2A | B | 3 | 2 | DNA-directed RNA polymerase II subunit RPB1 |
| CUL3 | C | 2 | 2 | E3 component, WNK degradation, BTB domain, cell cycle, cyclin E degradation |
| CDCA7L | D | 1 | 1 | Cell cycle, transcription co-activator, c-Myc interactor, FoxP3-binding |
| C10orf46 (CACUL1) | D | 1 | 1 | CDK2-associated, cell cycle, promotes proliferation |
| E2F2 | D | 1 | 1 | Cell cycle, transcription factor, T-cell quiescence |
| FAM65B | D | 1 | 1 | Skeletal muscle development, hearing |
| FLJ20105 (PICH) | D | 1 | 1 | cell division |
| HUWE1 | D | 6 | 1 | Ubiquitination and proteasomal degradation, Base-excision repair, neural differentiation and proliferation |
| KIF11 | D | 1 | 1 | ATP-dependent microtubule motor activity |
| PCBP1 | D | 2 | 1 | Cadherin binding, involved in cell-cell adhesion, Burkitt lymphoma |
| RPN2 | D | 1 | 1 | Endopeptidase activity, ubiquitin-dependent protein catabolic process |
| TCF12 | D | 1 | 1 | Immune response, regulation of transcription |
| ZBTB40 | D | 2 | 1 | Bone mineralization, cellular response to DNA damage stimulus |

Y2H assays were performed by Hybrigenics Services (Paris, France). The coding sequence for human ARMC5 cDNA (aa 30–935) served as bait to screen a random-primed human thymocyte cDNA library. Eighty million yeast clones (eightfold the complexity of the library) were screened. One hundred sixty-five His$^+$ colonies were selected. The prey fragments of positive clones were amplified by PCR and sequenced at their 5' and 3' junctions. The resulting sequences were used to identify corresponding interacting proteins in the GenBank database via a fully automated procedure. A Predicted Biological Confidence (PBC) score (from A to F; A being of very high confidence in the interaction and F being experimentally proven artefacts) was attributed to each interaction. Sixteen proteins with PBC scores between A and D are listed, if their coding sequences are in-frame and have no in-frame stop codons. Binding clones: number of total clones interacting with the bait. Different clones: number of different clones of the same cDNA interacting with the bait. Known functions of the prey proteins are described.

in-frame stop codons. A complete list of binding proteins identified by Y2H assay and a map showing the interaction regions between ARMC5 and its binding partners are provided in the Supplementary Materials section (Supplementary Data 1 and Supplementary Fig. 12).

## Discussion

Our study demonstrated that *Armc5* mRNA was highly expressed in the thymus and adrenal glands. Its deletion led to small body size in mice and compromised T-cell proliferation and differentiation. KO mice presented defective induction of EAE and anti-LCMV immune responses. KO mice developed adrenal gland hyperplasia in old age. ARMC5 is a protein without enzymatic activity. Our Y2H assays identified 16 candidate ARMC5-binding proteins potentially capable of linking ARMC5 to different signalling pathways involved in cell cycling and apoptosis.

*Armc5* expression at the mRNA level was upregulated immediately (within 2 h) after CD4 T-cell activation by T cell receptor (TCR) ligation and less so and at a slower pace in CD8 cells. Its expression level then declined in the following days (Fig. 1d). *Armc5* expression in CD4 cells cultured under Th1 or Th17 conditions after 1 day remained low (Supplementary Fig. 13), and was not influenced by the presence of different lymphokines, such as IL-2, IL-6 or TGF-β1 (Supplementary Fig. 14). *Armc5* mRNA expression in CD8 cells 8 days after LCMV infection was significantly lower than in naive CD8 cells (Supplementary Fig. 15). These data suggest that this molecule is probably important in the early stage of TCR-triggered T-cell activation to prepare cells for entry into the cell cycle. This notion is supported by cell cycle analysis, which revealed that KO T cells were compromised in G1/S progression (Fig. 4b).

We found reduced numbers of infiltrating T cells as well as Th1 (IFN-γ$^+$) and Th17 (IL-17$^+$) cells in the CNS of KO EAE mice compared to WT EAE controls. Such decreases were likely responsible for the diminished EAE manifestations in KO mice.

Reduced CNS lymphocyte infiltration could be caused by compromised clonal expansion/differentiation of T cells in the periphery, defective migration of such cells into the CNS, reduced expansion/differentiation of these cells in the CNS, decreased apoptosis of cells in the periphery and CNS, or all of the above. Defective KO T-cell clonal expansion/differentiation in the periphery was apparent according to our *in vitro* and *in vivo* results (Figs 4 and 7b), but whether this is also the case in the CNS remains to be studied.

We demonstrated that ARMC5 deletion resulted in comprised TCR-stimulated proliferation of both CD4$^+$ and CD8$^+$ T cells *in vitro* and LCMV-specific CD8$^+$ T-cell clonal expansion *in vivo*. Moreover, we observed a significant reduction in SLECs in KO mice following LCMV infection while MPECs were increased. Taken together, these results suggest a function for ARMC5 in promoting T-cell growth following TCR engagement possibly by regulating activation threshold levels; this provides a potential explanation for the observed increase in T-cell death following FasL engagement (Fig. 4c) in KO mice. It is possible that augmented apoptosis also plays a role in compromised Th1 and Th17 development from naive KO CD4 cells. However, *ARMC5* mutations lead to PMAH in humans and diffuse adrenal gland hyperplasia in mice (Fig. 3d), indicating that it has a default function of repressing adrenal cell proliferation, or a default pro-apoptotic function, or both. Indeed, an *in vitro* study of the human adrenal gland cell line H295R revealed that ARMC5 overexpression culminates in apoptosis[34], supporting its putative default anti-apoptotic function in adrenal glands. Dichotomous functions of ARMC5 in T cells versus adrenal glands indicate its tissue- or context-specificity, likely due to ARMC5's association with different binding partners. In different types of cells, ARMC5 might preferentially bind to a certain partner, depending on its relative abundance in a given cell type or cell status. Consequently, the default function of ARMC5 in certain types of cells or cells with a given status could be either pro- or anti-proliferation, pro- or anti-apoptosis, or neutral. It could explain the different phenotypes seen in T cells versus adrenal

glands, in terms of proliferation and apoptosis. It could also explain the obvious dilemma that KO T-cell development in the thymus, which involves fast thymocyte proliferation, is normal, but TCR-stimulated T-cell proliferation/differentiation and virus-induced T-cell clonal expansion are defective in KO mice.

Although B cells were not the focus of this study, we did demonstrate that KO B cells were compromised in proliferation triggered by BCR ligation. Although KO mice had normal serum IgG levels, it is possible that, under strenuous conditions, KO mice might manifest defective humoral immune responses.

Based on the functional results of our ARMC5 study, those from PMAH investigations, and protein association information from Y2H assays, we propose the following speculative model of ARMC5 mechanisms of action. ARMC5 transcription and protein expression are increased when the cells are activated. Induced ARMC5 forms dimers (or multimers) in cytosol. Such dimers are able to interact with different molecules in pathways regulating cell cycling and apoptosis, for example, CUL3 for cell cycling and DAPK1 for apoptosis. Therefore, depending on the relative abundance of binding proteins in different cell types and cells in different states, ARMC5 may interact preferentially with one or the other, leading to opposite functions in regulating cell proliferation and apoptosis. It should be noted that list of associating proteins might expand, pending further verification.

In summary, we demonstrated that ARMC5 has vital functions in fetal development, T-cell biology, immune responses and adrenal gland biology. We have created *Armc5* KO mice as the first animal model of a rare human disease: PMAH. Our mechanistic study to identify ARMC5-binding partners has laid the groundwork for further elucidation of ARMC5's mechanisms of action. With a better understanding of these mechanisms, this molecule may be deployed as a therapeutic target in immune and endocrine disorders.

## Methods

**ISH.** To localize *Armc5* mRNA, 1526-bp (starting from GATATC to the end) mouse *Armc5* cDNA (GenBank: BC032200, cDNA clone MGC: 36606) in pSPORT1 was employed as template for S and AS riboprobe synthesis, with SP6 and T7 RNA polymerase for both [35]S-UTP and [35]S-CTP incorporation[37].

Tissues from WT mice were frozen in $-35\,^{\circ}\mathrm{C}$ isopentane and kept at $-80\,^{\circ}\mathrm{C}$ until they were sectioned. ISH, X-ray and emulsion autoradiography focused on 10-μm thick cryostat-cut sections. Briefly, overnight hybridization at $55\,^{\circ}\mathrm{C}$ was followed by extensive washing and digestion with RNase to eliminate non-specifically bound probes. Anatomical level images of ISH were generated using X-Ray film autoradiography after 4 days' exposure. Microscopical level ISH was produced by dipping sections in NTB-2 photographic emulsion (Kodak). The exposure time was 28 days. The autoradiography labelling was revealed by D19 Developer (Kodak) and fixation with 35% sodium thiosulphate. Slides were left unstained or slightly stained with HE (ref. 37).

**RT-qPCR.** *Armc5* mRNA in thymocytes, T cells, B cells and tissues from KO and WT mice was measured by RT-qPCR. Total RNA was extracted with TRIzol (Invitrogen, Carlsbad, CA, USA) and then reverse-transcribed with Superscript II reverse-transcriptase (Invitrogen). Thymocytes were stained with anti-CD4 (1:400, Clone RM4-5, BD Bioscience), anti-CD8 (1:400, Clone 53-6.7, BioLegend), anti-CD25 (1:200, Clone PC61.5, eBioscience) and anti-CD44 (1:200, Clone IM7, BioLegend) Abs. CD4[+] cells, CD8[+] cells, DP cells, DN cells and DN cells in different stages were sorted by flow cytometry. T cells and B cells were isolated by magnetic beads (EasySep, Stem Cell Technology, Vancouver, BC, Canada). For RT-qPCR measurement of *Armc5* expression during T-cell activation, mouse T-activator CD3/CD28 Dynabeads (ThermoFisher Scientific, Burlington, ON, Canada) were used for T-cell activation *in vitro*, to avoid introducing Ag-presenting cells in purified CD4 or CD8 cells.

Forward and reverse primers were 5′-CAG TTA TGT GGT GAA GCT GGC GAA-3′ and 5′-ACC CTC AGA AAT CAG CCA CAA CCT-3′, respectively. A 139-bp product was detected with the following amplification program: $95\,^{\circ}\mathrm{C}\times 15\,\mathrm{min}$, 1 cycle; $95\,^{\circ}\mathrm{C}\times 10\,\mathrm{s}$, $59\,^{\circ}\mathrm{C}\times 15\,\mathrm{s}$, $72\,^{\circ}\mathrm{C}\times 25\,\mathrm{s}$, 35 cycles. *β-actin* mRNA levels were measured as internal controls. Forward and reverse primers were 5′-TCG TAC CAC AGG CAT TGT GAT GGA-3′ and 5′-TGA TGT CAC GCA CGA TTT CCC TCT-3′, respectively, with the same amplification program as

for *Armc5* mRNA. The data were expressed as ratios of *Armc5* versus *β-actin* signals.

**Armc5 overexpression in L cells.** L cells (ATCC CRL-2648, ATCC) were transiently transfected with pReceiver-Lv120 plasmid expressing mouse *Armc5* with HA tag (EX-Mm23477-LV120, GeneCopoeia, Rockville, MD, USA) for 2 days, and fixed with 4% paraformaldehyde. Subcellular Armc5 localization in L cells was detected by immunofluorescence with biotinylated rat anti-HA Ab (1:500, 12158167001, Roche, Laval, QC, Canada), followed by FITC-conjugated streptavidin (1:2,000, S11223, ThermoFisher, Burlington, ON, Canada). The L cells were not authenticated and possible mycoplasma contamination was not tested.

**Generation of Armc5 KO mice.** A PCR fragment amplified from the *Armc5* cDNA sequence served as probe to isolate genomic BAC DNA clone 7O8 from the RPCI-22 129/sv mouse BAC genomic library. The targeting vector was constructed by recombination and routine cloning methods, with a 15-kb *Armc5* genomic fragment from clone 7O8 as starting material. A 2.7-kb HindIII/EcoRV genomic fragment containing exon 1–3 was replaced by a 1.1-kb Neo cassette from pMC1Neo-Poly A flanked by two diagnostic restriction sites, EcoRV, and HindIII, as illustrated in Fig. 2a. The final targeting fragment was excised from its cloning vector backbone by NotI/EcoRI digestion and electroporated into R1 embryonic stem (ES) cells for G418 selection. Targeted ES cell clones were injected into C57BL/6J blastocysts. Chimeric male mice were mated with C57BL/6 females to establish mutated *Armc5* allele germline transmission.

Southern blotting with probes corresponding to 5′ and 3′ sequences outside the targeting region, as illustrated in Fig. 2a (red squares), screened for gene-targeted ES cells and eventually confirmed gene deletion in mouse tail DNA. With the 5′ probe, the targeted allele presented a 6.6-kb EcoRV band, and the WT allele, a 9.3-kb EcoRV band. With the 3′ probe, the targeted allele presented an 8.7-kb HindIII band, and the WT allele, a 12.5-kb HindIII band (Supplementary Fig. 3).

Heterozygous mice were backcrossed to the C57BL/6J background for eight generations and then crossed with 129/sv mice. WT and KO mice in the C57BL/6J × 129/sv F1 background were studied. All animals were housed under specific pathogen-free conditions and handled in accordance with a protocol approved by the Institutional Animal Protection Committees of the CRCHUM and INRS-IAF.

**Serum total IgG measurement.** Flat bottom 96-well plates (Costar EIA/RIA, No. 3369, Fisher Scientific) were coated with goat anti-mouse IgG (100 μl per well, $1\,\mu\mathrm{g}\,\mathrm{ml}^{-1}$ in phosphate-buffered saline (PBS)) and incubated overnight at $4\,^{\circ}\mathrm{C}$. After five times of washings with PBS containing 0.05% Tween 20, the plates were blocked with PBS containing 3% BSA and 5% fetal bovine serum for 1.5 h at room temperature. After five washings, diluted serum samples ($1{:}10^5$) and serially diluted standard mouse IgG (sc-2025, Santa Cruz) were added to the wells (100 μl per well) and incubated at $37\,^{\circ}\mathrm{C}$ for 1 h. The plates were then washed 10 times, and diluted (1:4,000) horse radish peroxidase-conjugated horse anti-mouse IgG (#7076 S, Cell Signaling Technology) was added (100 μl per well) to the wells. The plates were incubated for 1 h at $37\,^{\circ}\mathrm{C}$. After another 10 washings, 1-Step Ultra TMB-ELISA Substrate Solution (#34028, Thermo Scientific) was added to the wells (100 μl per well). The plates were incubated in the dark at room temperature for 20–30 min, and the reaction was stopped by 2 M sulfuric acid (100 μl per well). Optical density at 450 nm of reactants was measured. Samples were assayed in duplicate. Mouse total IgG concentrations were calculated according to a standard curve established by serial dilutions of standard mouse IgG. Assay sensitivity was in the 0.39 and $6.25\,\mathrm{ng}\,\mathrm{ml}^{-1}$ range.

**Enzyme-linked immunosorbent assay (ELISA).** Glucocorticoid levels in WT and KO mouse sera were quantified by ELISA, detecting mouse glucocorticoids according to the manufacturer's instructions (MBS028416, MyBioSource, San Diego, CA, USA).

**Flow cytometry.** Single cell suspensions from the thymus, spleen and LN were prepared and stained immediately or after culture with Abs against CD3 (1:200, Clone 145-2C11, BD Bioscience), CD4 (1:400, Clone RM4-5, BD Bioscience), CD8 (1:400, Clone 53-6.7, BioLegend), CD25 (1:200, Clone PC61.5, eBioscience), CD44 (1:200, Clone IM7, BioLegend), CD45.1 (1:200, Clone A20, BD Bioscience), CD45.2 (1:200, Clone 104, BD Bioscience), CD62L (1:200, Clone MEL-14, BD Bioscience), CD107a (1:200, Clone 1D4B, BD Bioscience), CD127 (1:200, A019D5, BioLegend), KLRG1 (1:200, Clone 2F1/KLRG1, BioLegend), Thy1.2 (1:1000, Clone 30-H12, BioLegend), B220 (1:200, Clone RA3-6B2, BD Bioscience), 7AAD (1:25, 51-68981E, BD Bioscience), Annexin-V (1:50, 550474, BD Bioscience). In some experiments, intracellular proteins, such as IFN-γ (1:200, Clone XMG1.2, BD Bioscience), IL-17 (1:200, Clone TC11-18H10, BD Bioscience), FoxP3 (1:200, Clone 150D, BioLegend), T-bet (1:200, Clone 4B10, BioLegend), RORγt (1:200, Clone B2D, eBioscience) and TNF-α (1:200, Clone MP6-XT22, BD Bioscience) and Granzyme B (1:200, Clone GB11, BioLegend), were detected after the cells

were pre-stained with Abs against cell surface Ag, permeabilized with BD Cytofix/Cytoperm solution (BD Biosciences), and then stained with Abs against intracellular Ag[38,39].

Flow cytometry was also employed to assess LCMV-specific T cells. The synthetic peptides gp$_{33-41}$: KAVYNFATC (LCMV-GP, H-2D$^b$), np$_{396-405}$: FQPQNGQFI (LCMV-NP, H-2D$^b$) and gp$_{276-286}$: SGVENPGGYCL (LCMV-GP, H-2D$^b$) were purchased from Sigma-Genosys (Oakville, ON, Canada). PE-gp$_{33-41}$, PE-np$_{396-405}$ and PE-gp$_{276-286}$ H-2D$^b$ tetrameric complexes were synthesized in-house and applied at 1/100 dilution[39]. These MHC-tetramers served to detect LCMV-specific CD8$^+$ T cells on day 8 post-LCMV infection. Briefly, splenocytes were first stained with PE-gp$_{33-41}$, PE-np$_{396-405}$ or PE-gp$_{276-286}$ tetramers for 30 min at 37 °C, directly followed by surface staining (CD3, CD8, CD44 and CD62L) and dead cell exclusion (7AAD) for another 20 min at 4 °C. The cells were then fixed with 1% paraformaldehyde and samples were analysed by flow cytometry.

For intracellular cytokine staining, 10$^6$ splenocytes from LCMV-infected mice were maintained for 5 h at 37 °C in RPMI-1640 with 10% FCS and 55 μg ml$^{-1}$ β-ME, supplemented with a final concentration of 50 U ml$^{-1}$ IL-2, 5 μg ml$^{-1}$ Brefeldin A, 2 μM Monensin, 2.5 μg ml$^{-1}$ FITC-labelled anti-mouse CD107a and 5 μM gp$_{33-41}$ or gp$_{61-80}$ GLNGPDIYKGVYQFKSVEFD (LCMV-GP, I-Ab) synthetic peptide from Sigma-Genosys. After ex vivo incubation, surface staining and cell viability were verified with anti-mouse CD8a, CD4 and CD62L mAbs and 7AAD. The cells were then fixed, permeabilized and stained with anti-mouse TNF-α, IFN-γ and Granzyme B mAbs. Cytokine-producing T cells were analysed by flow cytometry[40].

**Lymphocyte proliferation and apoptosis in vitro.** Spleen cells were loaded with carboxyfluorescein succinimidyl ester (CFSE: 5 μM for 5 min). After washing, they were stimulated with soluble hamster anti-mouse CD3ε mAb (clone 145-2C11, 2 μg ml$^{-1}$; BD Biosciences) for T-cell proliferation assays. This protocol allows best long-term T-cell proliferation over a 4-day period, for clear demonstration of multiple cell proliferation rounds according to CFSE staining. In B-cell proliferation assays, CFSE-loaded Spleen cells were stimulated with goat anti-mouse IgM (5 μg ml$^{-1}$; Jackson ImmunoResearch), IL-4 (10 ng ml$^{-1}$) and goat-anti-mouse CD40 (2 μg ml$^{-1}$, Jackson ImmunoResearch). After 3–4 days, the cells were gated on CD4$^-$, CD8$^-$ or B220-positive cells, and their CFSE intensity was ascertained by flow cytometry.

To assess Th1 and Th17 cell proliferation, naive CD4 cells were loaded with CFSE, and cultured under Th1 and Th17 conditions for 4 days (detailed below). CD4$^+$ or intracellular IFN-γ$^+$ or IL-17$^+$ cells were then gated, and their CFSE intensity was assessed by flow cytometry.

For cell cycle analysis, total spleen cells were stimulated with anti-CD3ε mAb, as described above, and stained with anti-Thy1.2 mAb and propidium iodide (20 μg ml$^{-1}$) on days 0, 1 and 2. Thy1.2$^+$ T cells were gated and their propidium iodide signal strength was measured by flow cytometry.

For T-cell apoptosis analysis, spleen cells were stimulated with anti-CD3ε mAb (2 μg ml$^{-1}$) plus crosslinked human FasL-FLAG (0.133 μg ml$^{-1}$; FasL-FLAG was pre-incubated for 24 h at 4 °C at a 1:1 ratio with 0.133 μg ml$^{-1}$ mouse monoclonal Ab against FLAG (F1804, Sigma); the final concentration of crosslinked FasL-FLAG for culture was 0.6 μg ml$^{-1}$ (ref. 41)) and cultured for 4 h. Cells positive for CD4 or CD8 were gated and analysed for annexin V expression.

**Th1 and Th17 cell differentiation in vitro.** T-cell differentiation in vitro was undertaken as follows[38,39]. Naive CD4$^+$ T cells (CD4$^+$ CD62L$^+$CD44$^{low}$) were isolated from KO or WT mouse Spleen with EasySep mouse naive CD4$^+$ T-cell isolation kits (19765, Stem Cell Technology). Naive CD4$^+$ T cells from WT and KO mice (0.1 × 10$^6$ cells per well) were mixed with feeder cells (0.5 × 10$^6$ cells per well) and cultured in 96-well plates in the presence of soluble anti-CD3ε Ab (2 μg ml$^{-1}$). Feeder cells plus anti CD3ε Ab were used, as in our hands, they achieved the most consistent Th1 and Th17 differentiation conditions. Cultures were supplemented with recombinant mouse IL-12 (10 ng ml$^{-1}$; 419-ML, R&D Systems, Minneapolis, MN, USA) and anti-IL-4 mAb (5 μg ml$^{-1}$; MAB404, R&D Systems) for the Th1 condition, with recombinant mouse IL-6 (20 ng ml$^{-1}$; 406-ML, R&D Systems), recombinant human TGF-β1 (5 ng ml$^{-1}$; 240-B, R&D Systems) and anti-IL-4 (5 μg ml$^{-1}$) and anti-IFN-γ mAbs (5 μg ml$^{-1}$; MAB485, R&D Systems) for the Th17 condition. The cells were stimulated with PMA (10 μM) and ionomycin (100 μg ml$^{-1}$) in the presence of 5 μg ml$^{-1}$ Brefeldin A for the last 4 h of culture, and their intracellular IFN-γ, T-bet, IL-17 and RORγt were analysed by flow cytometry.

**Chimera generation.** Eight- to 10-week-old C57BL/6J (CD45.2$^+$) × C57B6.SJL (CD45.1$^+$) F1 mice were irradiated at 1,100 rads. Twenty-four hours later, they received i.v. 2 × 10$^6$ fetal liver cells from WT or KO mice) in the C57BL/6J × 129/sv (CD45.2$^+$) F1 background. Peripheral white blood cells of recipients were analysed by flow cytometry 8 weeks after fetal liver cell transplantation. Twelve weeks after transplantation, chimeras with successful implantation of donor-derived white blood cells were studied in in vitro T-cell function experiments and for EAE induction.

**EAE induction and assessment.** EAE was induced in 8- to 12-week-old female WT and KO mice[42]. Briefly, mice were immunized with 300 μg MOG$_{35-55}$ peptide (Biomatik, Wilmington, DE, USA) emulsified in complete Freund's adjuvant, followed by i.p. injection of 400 ng pertussis toxin (List Biological Laboratories, Campbell, CA, USA) on days 0 and 2. EAE development was scored daily between days 0 and 35 according to a scale ranging from 0 to 5, as follows: 0, no sign of paralysis, 1, weak tail; 2, paralysed tail; 3, paralysed tail and weakness of hind limbs; 4, completely paralysed hind limbs; 5, moribund. Scores were assigned in 0.5 unit increments when symptoms fell between 2 full scores.

Female chimeras with successful donor cell implantation (verified according to CD45.2 single-positive cells in peripheral blood) 12 weeks after fetal liver transplantation were also used for EAE induction. The same protocol described above was followed, except that 200 μg MOG$_{35-55}$ peptide for immunization and 200 ng pertussis toxin/injection were administered to each chimeric mouse.

**EAE histology.** To assess the degree of inflammation and CNS demyelination, EAE mice were killed on day 30 and perfused by intra-cardiac injection of PBS. Spinal cord sections were stained with H/E or NovaUltra Luxol Fast Blue Staining Kit (IHC World, Woodstock, MD, USA), according to the manufacturer's instructions. Each SC section was subdivided into four regions: 1 anterior, 1 posterior and 2 lateral. Each region was scored on a scale of 0-3 for lymphocyte infiltration and demyelination in a one-way blinded fashion. Thus, each animal had a potentially maximal score of 12 points for lymphocyte infiltration and demyelination, respectively[43,44]. Total pathological scores were the sum of these 2 parameters.

**Isolation of mononuclear cells from the spinal cord and brain.** Peripheral blood was removed from the spinal cord and brain by intra-cardiac perfusion through the left ventricle with heparinized ice-cold PBS. The spinal cord and brain were harvested, ground and then passed through a 70-μm mesh screen. The cells were centrifuged through a 40–90% discontinuous Percoll gradient. Mononuclear cells at the 40–60% Percoll interface were collected and stained for phenotype analysis.

**Differentiation/characterization of mouse Th1 and Th17 cells.** Cells from draining LN and mononuclear cells from the spinal cord and brain were further stimulated with PMA (5 nM) and ionomycin (500 ng ml$^{-1}$) for 4 h in the presence of Golgi Stop (554724, BD Biosciences), before being harvested. They were stained with Abs against cell surface Ag, fixed with Cytofix/Cytoperm solution (555028, BD Biosciences), and then stained with mAbs against intracellular IFN-γ (1:200, Clone XMG1.2, BD Bioscience) and IL-17 (1:200, Clone TC11-18H10, BD Bioscience). Stained cells were analysed by flow cytometry.

**LCMV infection.** LCMV-WE was obtained from Dr R.M. Zinkernagel (University of Zurich, Zurich, Switzerland). Viral stock was propagated in vitro, and viral titers were quantified by focus-forming assay[40]. Mice were infected by the i.v. route with 200 focus-forming units (ffu) of LCMV-WE. They were killed 8 days post-infection, and their spleens were harvested for primary immune response analysis. CD8$^+$ T cells were isolated from the spleen of naive or infected WT mice, with EasySep mouse CD8$^+$ T-cell isolation kits (19853, Stem Cell Technology). RNA from isolated cells was extracted with TRIzol (Invitrogen), followed by RT-qPCR.

**Y2H assay.** Y2H screening was performed by Hybrigenics Services (Paris, France). The coding sequence for human ARMC5 cDNA (aa 30–935) (GenBank accession number GI: 157426855) was PCR-amplified and cloned into pB29 as a N-terminal fusion protein to LexA (N-ARMC5-LexA-C). The construct was verified by sequencing the entire insert and served as bait to screen a random-primed human thymocyte cDNA library constructed in the pP6 plasmid. pB29 and pP6 vectors were derived from the original pBTM116 (refs 45,46) and pGADGH[47] plasmids, respectively.

Eighty million yeast clones (eightfold the complexity of the library) were screened via a mating approach with YHGX13 (Y187 ade2-10: loxP-kanMX-loxP, matα) and L40Gal4 (mata) yeast strains[48]. One hundred sixty-five His$^+$ colonies were selected on medium lacking tryptophan, leucine and histidine, and supplemented with 5.0 mM of 3-aminotriazole to quench bait auto-activation. Prey fragments of positive clones were amplified by PCR and sequenced at their 5′ and 3′ junctions. The resulting sequences were considered to identify corresponding interacting proteins in the GenBank database via a fully automated procedure. A Predicted Biological Confidence score was attributed to each interaction[36].

**Statistics.** For in vivo animal studies, the sample size was determined by estimation, based on our past experience and literature. No formal randomization was used, but littermates or age- and sex-matched WT and KO mice were used.

In general, two-tailed Student's *t*-tests were used. Chi-square test was used to compare the difference between two proportions. One-way analysis of variance followed with Bonferroni's multiple comparisons test was used in data from more than three groups. For the histology experiments, one-way blind examination was performed.

**Data availability.** The data that support the findings of this study are available in the text and the Supplementary Data files, or from the corresponding author on request.

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

## Acknowledgements

This work was supported by grants from the Canadian Institutes of Health Research to J.W. (MOP69089 and MOP 123389), A.L. (MOP89797) and H.L. (MOP97829). It was also funded by grants from the Natural Sciences and Engineering Research Council of Canada (203906-2012), the Juvenile Diabetes Research Foundation (17-2013-440), Fonds de recherche du Quebéc-Santé (Ag-06) to J.W., and the Jean-Louis Levesque Foundation

to J.W and A.L. The authors thank Dr M. Sarfati and her group for sorting thymocytes by flow cytometry.

**Author contributions**

Y.H., L.L., J.M., A.L. and J.W. conceived and designed the experiments. Y.H., L.L., J.M., W.J., H.L., T.C., S.Q., J.P. and M.M.M. performed the experiments. Y.H., L.L., J.M., W.J., H.L., T.C., M.M.M., B.H. and J.W. analysed the data. Y.H., L.L., A.L. and J.W. wrote the manuscript.

**Additional information**

**Competing financial interests:** The authors declare no competing financial interests.

