## [Peer Review File · Nature Communications]

Reviewers' comments:

Reviewer #1 (Remarks to the Author):

In this manuscript Hu et al describe the roles of ARMC5, a protein with unknown function, in development and T cell responses. This is significant as ARMC5 has been implicated in two diseases, primary macronodular adreanal hyperplasia and Cushing's syndrome. The authors report the generation of new ARMC5-KO mouse model, which they characterize for the first time. The authors show that ARMC5 is expressed in T cells and ARMC5-deficiency leads to defects in T cell proliferation and function. Furthermore, the authors propose a model how ARMC5 might function.

Overall this is an interesting study that could open new avenues to understanding the role of ARMC5, and it appears clear that ARMC5 plays important roles in T cells. However, in the current state, the experiments are not clear and the conclusions not sufficiently supported by the data.

The authors should address the following major points before the manuscript can be properly assessed:

1. The characterization of ARMC5 expression in T cells is insufficient. In Fig. 1D for example, ARMC5-transcript levels are shown over the course of 6 hours post activation. This is not enough; during this short period the cells don't even start proliferating. The authors should show expression data also at later time points, at least day 1, 2 and 3 post activation. Given their later experiments, it would also be important to show expression in Th1, Th17 and CD8 T cells. Are there differences in different T cell populations? The authors should also measure ARMC5 in naïve, memory and effector phenotype cells sorted from mice. This should be contrasted to another lymphocyte population such as B cells (with and without activation).
2. The authors have developed a new ARMC5 antibody. However, the only experiments in which they use it, is shown in Fig. 2D, which is rather unconvincing. I would expect to see a Western Blot that could back up RNA expression data. I also think the authors should make use of the intracellular stain and test if ARMC5 is expressed in Th1 and CD8 T cells during EAE or LCMV infection respectively.
3. The characterization of the proliferation defect in ARMC5-KO T cells is insufficient. Fig. 5a,b suggests that the KO cells don't proliferate at all. Is that correct. The experiment should be taken to a later time point to show stronger CFSE dilution in the WT and measure what happens in the KO. Is the defect in cytokine production due to a total lack of proliferation? Ideally, IFN γ and IL-17 should be shown versus CFSE dilution.
4. The authors should show their experiments related to apoptosis. They are discussed but data are not displayed.
5. The authors propose that ARMC5 regulates cell cycle. Is that the case? Can they show BrdU incorporation data for their in vivo experiments, or more convincing in vitro data?
6. Relating to Fig. 8, the authors need to show some primary data, such as flow cytometry of their tetramer staining and a phenotypic analysis of the antigen-specific cells (showing informative markers such as CD62L, CD44, PD1, KLRG1).

7. Importantly, the authors performed a yeast-2-hybrid assay to identify binding partners of ARMT5. This is important and apparently the basis of the model provided in Fig. 9. However, they don't provide any primary data related to this assay, and don't even mention which genes were identified and with which confidence, making it impossible to judge the quality of the data.

Minor points

1. The authors seem to conclude that T cells are specifically affected. If that is the case, are B cell responses in the mice normal? I understand that not everything can be analysed in the first report; however, it would be useful to test if Tfh cells, germinal centres and plasma cells develop in the LCMV model. This could easily be done by flow.
2. Do Tregs develop normally?
3. What happens when ARMC5 is overexpressed in T cells. The authors appear to have an over-expression construct (as shown in Fig. 1). Could they attempt to express it in T cells?
4. Could the authors test whether cytokines such as IL-2, IL-12 etc play any role in ARMT5 induction.
5. The manuscript should be carefully proof read. It contains many smaller errors. Also, the introduction could be improved.
6. The quality of some figure panels in Fig. 1 is poor (for example 1E). This should be improved. Also, the description of the figures in the legends is not always sufficient.
7. The model in Fig. 9 appears very speculative

Reviewer #2 (Remarks to the Author):

The paper by Wu reports novel findings on the function of ARMC5 a gene with hitherto unclear function, although somatic mutations in ARMC5 has recently been associated with macronodular adrenal hyperplasia and Cushing's syndrome. The authors have performed a comprehensive study of expression and characterized the consequences of knocking out the gene in mice. Unfortunately the KO model displayed reduced fertility, but an alternative approach involving transplanting of fetal liver from KO-mice to fatally irradiated recipients enabled the authors to study the effects of ARMC5. They show that ARMC5 regulate T cell proliferation and modulates disease progression in EAE. Taken together the data adds valuable information to the function of ARMC5.

Many different technologies and approaches have been used. The paper is well written and the arguments are easy to follow. The conclusions are valid. An extensive panel of figures. Some of them are not needed to follow the flow of arguments and could be moved to a supplemental, others would benefit from revision (detailed below).

Specific comments

1. ARMC5 expression in KO. Fig 2D shows that ARMC5 protein expression is deleted in KO thymocytes, but I do not understand how the figure shows protein deletion in KO. What does WT

62.7 and KO 48.8 percent mean.

2. The expression of ARMC5 was higher in the thymic cortex than in the medulla. Was there any expression in medullary thymic epithelial cells or was the expression located only in thymocytes?
3. Assessment of adrenal function. The authors present histology and glucocorticoid levels in WT and KO mice. It is unclear what the kit they use measure. The active glucocorticoid of mice is corticosterone, is the kit measuring corticosterone or a mixture of steroids? When were the mice sampled? Corticosterone in mice show circadian variation opposite of humans since they are nocturnal animals. The levels presented in Fig 3F seems to quite low, are these morning values? If so it might be difficult to find a difference and also detect adrenocortical insufficiency.
4. Since the number of panels and figures in the paper are quite high, Fig 3D-F, Figure 4A-D and Fig 6A can be moved to a Supplemental as the findings are very similar in WT and KO.
5. Fig 7E label I in IL-17 is missing
6. The concept figure (Fig.9) has no legend - this should be provided. The mechanism of action described in the Discussion can be improved and focused. As it stands it is rather diffuse.
7. Spelling
 - a. Remove etc in introduction
 - b. Misspelling in last paragraph on page 4 that instead of than

Reviewer #3 (Remarks to the Author):

In this manuscript, Hu et al. explore potential roles for the ARM domain-containing protein, Armc5. Through the use of a knockout strategy in combination with descriptive and functional assays, the authors find that mice lacking Armc5 have severe developmental defects and ineffective immune responses. In particular, the authors make the novel finding that T cell responses, both CD4+ and CD8+, appear compromised, as these cells display defects in both proliferation and effector functions. Interestingly, T cell populations in knockout mice are neither sufficient to support the complete onset of the autoimmune disease EAE, nor able to efficiently clear an acute LCMV infection. Finally, the authors utilize a yeast two-hybrid approach in an attempt to gain insight into the Armc5-dependent mechanisms that may be responsible for the observed defects. Overall, the authors present some intriguing and novel findings. However, the manuscript is largely descriptive and the absence of mechanistic data diminishes the significance and impact of the work in its current form. I have the following comments and/or concerns:

1. An immunoblot should be performed to examine Armc5 protein in wild-type vs. Armc5^{-/-} tissues. The data from Figure 2D are unimpressive as there is little difference between WT and KO. As such, it is particularly important that the antibody used for flow cytometry be validated by immunoblot. Also, it is difficult to assess the flow cytometry data without an isotype control being displayed. The use of an isotype control is critical. It should be shown.
2. The defects in T cell proliferation in Figures 5A and 5B are quite striking. However, it appears that only one cell division has been measured. A more extensive analysis may be interesting in that it

would allow the authors to determine whether the block is complete or whether there is a more subtle delay in cell proliferation. A more detailed cell cycle analysis may also provide further information as to the stage of division in which the block occurs.

3. The authors alternate between using feeder cells in combination with anti-CD3 and anti-CD3/anti-CD28 in a number of experiments (Figs. 5A, 5B, 5C, 6B, 6C). It is difficult to discern why the authors chose to do this and which experiments were performed with each specific set of conditions. Were differences observed in experimental outcome depending on the particular method of cell stimulation? The authors should clearly state why alternating conditions were used and the condition that was chosen for a particular experiment.

4. As with the defects in proliferation, the T cell differentiation experiments are very intriguing. However, outside of the analysis of Ifng and IL-17 expression, we know very little with regards to the mechanism resulting in the defect. Is the defect specific to cytokine (Ifng, IL-17) secretion? What is the expression of T-bet and ROR gamma T? Do these cells express cytokine receptors consistent with Th1 and Th17 development? The impact of the current data would be significantly enhanced with an attempt to further elucidate the mechanism behind the developmental defect.

5. What are the dynamics of Armc5 expression during T cell development? Outside of figure 1D, there is no attempt to examine Armc5 expression (particularly at the protein level) during T cell proliferation or activation. Expression patterns should be assessed as these data could provide insight into potential mechanistic roles for Armc5 in T cell differentiation.

6. There is not sufficient data to justify the model proposed in Figure 9.

Minor concerns

1. No Th0 control for the Th17 cells is shown in Figure 6D.

2. The assertion that the immune response to LCMV is largely CD8+ cell mediated is overstated. LCMV has been used extensively to study CD4+ T cell responses. As such, the CD4+ response in the LCMV studies (Figure 8) may be an area of interest for the authors to explore.

3. The image in Figure 1E is very dark and thus may be difficult for the audience to visualize. Also, as this is human ARMC5, does murine Armc5 similarly display cytosolic localization?

Response to referees

Point-to-point replies

The original comments are in italic.

Reviewer #1

In this manuscript Hu et al describe the roles of ARMC5, a protein with unknown function, in development and T cell responses. This is significant as ARMC5 has been implicated in two diseases, primary macronodular adrenal hyperplasia and Cushing's syndrome. The authors report the generation of new ARMC5-KO mouse model, which they characterize for the first time. The authors show that ARMC5 is expressed in T cells and ARMC5-deficiency leads to defects in T cell proliferation and function. Furthermore, the authors propose a model how ARMC5 might function.

Overall this is an interesting study that could open new avenues to understanding the role of ARMC5, and it appears clear that ARMT5 plays important roles in T cells. However, in the current state, the experiments are not clear and the conclusions not sufficiently supported by the data.

The authors should address the following major points before the manuscript can be properly assessed:

1. The characterization of ARMC5 expression in T cells is insufficient. In Fig. 1d for example, ARMC5-transcript levels are shown over the course of 6 hours post activation. This is not enough; during this short period the cells don't even start proliferating. The authors should show expression data also at later time points, at least day1, 2 and 3 post activation. Given their later experiments, it would also be important to show expression in Th1, Th17 and CD8 T cells. Are there differences in different T cell populations? The authors should also measure ARMC5 in naïve, memory and effector phenotype cells sorted from mice. This should be contrasted to another lymphocyte population such as B cells (with and without activation).

We conducted pilot study to determine the expression kinetics of ARMC5 during T cell activation, and the increase only occurs within 6 h. That is why we only presented the 0-6 h data. After the initial peak expression at 3-5 h, the expression decreases and remains low until 72 h. This early rise of ARMC5 expression suggests that this molecule is required for proper T cell activation in the G0-G1 phase, even

though the cell division has not started at this stage. We have now presented the ARMC5 mRNA expression of CD4 and CD8 cells during 1-16 h and 24-72 h in Fig. 1d. The ARMC5 expression for Th1 and Th17 cells within the long-term time frame (24, 28 and 72 h) are also presented (S. Fig. 11). As requested, ARMC5 expression in naïve and T_{mem} (S. Fig. 1d) and during the course of B cell activation (S. Fig. 14).

2. The authors have developed a new ARMC5 antibody. However, the only experiments in which they use it, is shown in Fig. 2d, which is rather unconvincing. I would expect to see a Western Blot that could back up RNA expression data. I also think the authors should make use of the intracellular stain and test if ARMC5 is expressed in Th1 and CD8 T cells during EAE or LCMV infection respectively.

None of the commercial Abs (we tested all of them) can detect specifically the ARMC5 protein by Western or flow cytometry. The previous publications by other groups measuring ARMC5 proteins by Western are likely reporting non-specific bands. Our own Abs could not be used for Western. When it is used in flow cytometry, it is slightly better than the commercial Ab, but as the reviewer pointed out, it is not very convincing. In view of such technical limitations, we decided to remove our original flow cytometry histogram and rely on RT-PCR and Southern to prove the deletion of ARMC5 in the KO mice. Based on RT-PCR and Southern, as well as the prominent phenotypes in the body size, T cells and adrenal glands of the KO mice, we hope the reviewer agree that there is little doubt that ARMC5 is deleted. We have also relied on RT-PCR to determine the ARMC5 expression levels in various stages of T cell activation, and in T cell subpopulations, and thymocyte subpopulations. As per reviewer's critique, we now presented ARMC5 expression levels in CD8 cells on day 8 post LCMV infection (S. Fig. 12). With regard to ARMC5 expression in Th1 cells, we presented data showing ARMC5 expression one days 1, 2 and 3 during the course of Th1 differentiation in vitro (S. Fig. 11).

3. The characterization of the proliferation defect in ARMC5-KO T cells is insufficient. Fig. 5a, b suggests that the KO cells don't proliferate at all. Is that correct? The experiment should be taken to a later time point to show stronger CFSE dilution in the WT and measure what happens in the KO. Is the defect in cytokine production due to a total lack of proliferation? Ideally, IFN γ and IL-17 should be shown versus CFSE dilution.

We now show proliferation history of 4 days for CD4, CD8 (Fig. 4a), Th1 (IFN γ ⁺) and Th17 cells (IL-17⁺) (Fig. 5a) of KO and WT mice, using CFSE. The KO T cells do proliferate, but at a reduced rate. We also showed that KO Th1 and Th17 cells had a history of reduced proliferation according to CFSE

staining (Fig. 5a). Thus, the Th1- and Th17-secreting cell numbers are reduced in KO mice due to reduced proliferation, as evidenced by the CFSE data. The reduction is also caused by compromised differentiation, as the reduced Th1 and Th17 cell percentage is based on the ratios of intracellular Th1 and Th17-positive CD4 cells among total CD4 cells, which have already been proliferated; this reduced percentage reflects defects of CD4 differentiation

4. The authors should show their experiments related to apoptosis. They are discussed but data are not displayed.

We have now presented results about FasL-induced apoptosis in T cells. The KO T cells showed increased apoptosis and this result is now shown in Fig. 4c.

5. The authors propose that ARMT5 regulates cell cycle. Is that the case? Can they show Brdu incorporation data for their in vivo experiments, or more convincing in vitro data?

Brdu experiments *in vivo* will only show the proliferation history but not the defects in a certain phase of cell cycle. We did cell cycle analysis *in vitro*, and showed that KO T cells are compromised in G1/S-phase entry (Fig. 4b). As primary spleen T cells are difficult to be blocked and released at the G2/M phase, so we could not test whether ARMC5 has a function in G2/M phase entry. However, most cell cycle regulation events occur at the G1/S phase boundary.

6. Relating to Fig. 8, the authors need to show some primary data, such as flow cytometry of their tetramer staining and a phenotypic analysis of the antigen-specific cells (showing informative markers such as CD62L, CD44, PD1, KLRG1).

As per reviewer's suggestion, we now show representative histograms of tetramer staining, and CD62L, CD44, PD1, KLRG1 staining in supplementary data (S. Fig. 9).

7. Importantly, the authors performed a yeast-2-hybrid assay to identify binding partners of ARMT5. This is important and apparently the basis of the model provided in Fig. 9. However, they don't provide any primary data related to this assay, and don't even mention which genes were identified and with which confidence, making it impossible to judge the quality of the data.

We have now presented the primary data of Y2H in the supplementary materials (S. Table 1 and S. Fig. 10), which provide detailed information of all the hits and their interaction regions with ARMC5.

Minor points

1. The authors seem to conclude that T cells are specifically affected. If that is the case, are B cell responses in the mice normal? I understand that not everything can be analyzed in the first report; however, it would be useful to test if Tfh cells, germinal centers and plasma cells develop in the LCMV model. This could easily be done by flow.

We thank reviewer's understanding. Indeed, this is first report of the KO mice and first detailed study of ARMC5's roles in development and T cell function. The effect of ARMC5 on B cells is not really within the scope of our study. However, we understand the curiosity of the reviewer, and conducted some additional experiments related to the function of B cells. We showed that KO B cell proliferation is compromised (Fig. 4a). Tfh cell population in the naïve KO spleen has no significant change (S. Fig. 6), and there is no apparent defect in the appearance of germinal centers in the lymph nodes (S. Fig. 7), nor levels of total serum IgG (S. Fig. 8). Therefore, we could conclude that under an unchallenged condition, the humoral immune response of the KO mice is in the normal range. Detailed study on the humoral immunity of the KO mice under pathological conditions is in our agenda and will be reported elsewhere.

2. Do Tregs develop normally?

This is a question we prefer to address in a separate publication, but we provide the reviewer with some answers herewith, and the data are shown in Figs. 1 and 2 For Reviewers. The percentage of natural Treg from unmanipulated KO mice and Treg from EAE KO mice are comparable to those of WT mice. This means that the reduced EAE induction is not due to increased Treg. This is now mentioned in the text but marked as "data not shown". There is no consistent change in in vitro differentiation of Treg from naïve KO CD4 cells, but sometimes we see an increase of FoxP3+CD25+ cells, accompanied by an increase of ARMC5 expression at the late stage of Treg differentiation. We suspect ARMC5 might have something to do with FoxP3 degradation, but the physiological and pathophysiological relevance of such observation is not clear at the time being. We are actively working on this aspect of ARMC5 and prefer to report our findings once we have a better and mechanistic standing of the issue. We hope the reviewer would agree with us for not dwelling on Treg in this current publication.

Figure 1 for reviewers. Normal levels of Treg cells in unimmunized chimeric KO mice

Percentages of CD25⁺FoxP3⁺-positive cells among spleen CD4 from derived from KO or WT donor liver cells were determined by flow cytometry. The experiments were conducted more than 3 times and representative histograms are shown.

Figure 2 for reviewers. Normal levels of Treg cells in the draining LN of KO mice on day 17 during EAE induction

LN cells and central nervous system (CNS)-infiltrating cells from WT and WT mice on day 18 after EAE induction were harvested, and their CD4 cells were analyzed for CD25⁺FoxP3⁺ expression by flow cytometry. The percentages of CD25⁺FoxP3⁺ cells among CD4 cells are indicated. Means + SD of percentages of Treg cells among CD4 cells are shown. Mouse numbers (n) are indicated.

3. *What happens when ARMC5 is overexpressed in T cells. The authors appear to have an over-expression construct (as shown in Fig. 1). Could they attempt to express it in T cells?*

In Figure 1, we used an expression plasmid to overexpress ARMC5 in L cells, which are adherent cells and are easy to be transfected by plasmids. However, this approach will not work for T cells, which will need lentivirus-based vector to overexpress exogenous genes. Therefore, we do not have a ready tool reagent to overexpress ARMC5 in T cells. As we now know, the phenotype of a cell, organ or an animal is not simply controlled by the qualitative expression of a gene, but also by the quantity of the gene expression. Therefore, there is no telling what the phenotype of cells overexpressing ARMC5 will be, and

the overexpressing results might or might not corroborate the KO results. Certainly, to discover the phenotype of ARMC5 overexpressing is an interesting project, but our current work addresses what happens if it is deleted. The reviewer's point is well taken, and we are certainly very much interested in reporting in the future about what happens if ARMC5 is overexpressed.

4. Could the authors test whether cytokines such as IL-2, IL-12 etc play any role in ARMT5 induction.

We have now provided additional data in the supplementary materials (S. Fig. 12) about ARMC5 expression when T cells are cultured with different cytokines. Basically, the ARMC5 expression levels are reduced overtime, and we do not see any stimulation by different cytokines tested so far. So, ARMC5 expression is not responsive to cytokine but only gets a quick upregulation followed by a decline when TCR signaling is triggered.

5. The manuscript should be carefully proof read. It contains many smaller errors. Also, the introduction could be improved.

The manuscript has been carefully proof read, and the introduction has been improved, as suggested by the reviewer.

6. The quality of some figure panels in Fig. 1 is poor (for example 1E). This should be improved. Also, the description of the figures in the legends is not always sufficient.

We have improved the contrast of Fig. 1. It is probably due to the display and resolution of the figures in the computer screen. More details are given in the figure legends.

7. The model in Fig. 9 appears very speculative.

We concur that the model is speculative at this time, but feel that the diagram illustrates a general principle of how ARMC5 works. We suggest keeping this figure but moving it to the supplementary data to help the readers to grasp the concept. We now emphasize in the legend that this model is speculative and the exact ARMC5-binding molecules in the graph are subjected to change pending on further confirmatory studies.

Of course, if the review insists to remove this figure, we will comply.

Reviewer #2

The paper by Wu reports novel findings on the function of ARMC5 a gene with hitherto unclear function, although somatic mutations in ARMC5 has recently been associated with macronodular adrenal hyperplasia and Cushing's syndrome. The authors have performed a comprehensive study of expression and characterized the consequences of knocking out the gene in mice. Unfortunately, the KO model displayed reduced fertility, but an alternative approach involving transplanting of fetal liver from KO-mice to fatally irradiated recipients enabled the authors to study the effects of ARMC5. They show that ARMC5 regulate T cell proliferation and modulates disease progression in EAE. Taken together the data adds valuable information to the function of ARMC5.

Many different technologies and approaches have been used. The paper is well written and the arguments are easy to follow. The conclusions are valid. An extensive panel of figures. Some of them are not needed to follow the flow of arguments and could be moved to a supplemental, others would benefit from revision (detailed below).

Specific comments

1. ARMC5 expression in KO. Fig 2d shows that ARMC5 protein expression is deleted in KO thymocytes, but I do not understand how the figure shows protein deletion in KO. What does WT 62.7 and KO 48.8 percent mean.

The value is the mean fluorescent intensity. It means that the KO T cells (5000 cells) have lower collective intensity of the labeling. With that said, we concur with other reviewers that the data is not very convincing. Currently, there is no ARMC5-specific Abs (we test almost all the commercial Abs, but none of them is specific to ARMC5); and ours is better but is still not of sufficient specificity, with high background. In view of such technical limitations, we decided to remove our original flow cytometry histogram in Fig. 2d, and rely on RT-PCR and Southern to prove the deletion of ARMC5 in the KO mice. Based on RT-PCR and Southern, as well as the prominent phenotypes in the body size, T cells and adrenal glands of the KO mice, we hope the reviewer agree that there is little doubt that ARMC5 is deleted.

2. The expression of ARMC5 was higher in the thymic cortex than in the medulla. Was there any expression in medullary thymic epithelial cells or was the expression located only in thymocytes?

Thymic cortex has much higher cell density than medulla. That is why we see stronger ARMC5 signals in the cortex than in medulla. As a matter of fact, Fig. 1b-III shows clearly the different cell density in the cortex versus medulla. We have now compared the ARMC5 expression in the thymocytes and that in the thymic stromal cells (most of them epithelial cells), and showed that ARMC5 expression is comparable in these two types of cells (S. Fig. 1a).

3. Assessment of adrenal function. The authors present histology and glucocorticoid levels in WT and KO mice. It is unclear what the kit they use measure. The active glucocorticoid of mice is corticosterone, is the kit measuring corticosterone or a mixture of steroids? When were the mice sampled? Corticosterone in mice show circadian variation opposite of humans since they are nocturnal animals. The levels presented in Fig 3f seems to quite low, are these morning values? If so it might be difficult to find a difference and also detect adrenocortical insufficiency.

The kit we are using measures a mixture of glucocorticoids (not all the steroids), which in mice is mainly corticosterone. The blood of the mice was collected between 12:30 and 1:30 pm. In humans, ARMC5 mutation is associated with Cushing's syndrome. To diagnose Cushing's syndrome, the patient saliva is normally collected in the late evening around 9-10 pm, when elevated glucocorticoids secretion, if there is any, can be easily detected, because the background is low and background variation due to activities is small. This time frame in human clinical tests is equivalent to the late morning in mice. We intend to measure whether there is glucocorticoid over-production in KO mice, as is the case in humans. So, the blood has to be collected at the nadir of the secretion but not the peak of the secretion. In the revised manuscript, we have introduced very interesting new data, showing that aged ARMC5 mice (15-20 months old) presented adrenal gland hyperplasia, accompanied by elevated serum glucocorticoid levels (Figs. 3d and 3e), reminiscent of primary macronodular adrenal gland hyperplasia (PMAH) in aged humans with ARMC5 mutations. The increase in the glucocorticoid levels in the KO mice is significant although moderate. This is also the case in PMAH, in which the increase is not florid and is due to the large adrenal gland mass but not due to increased secretion. As a matter fact, the glucocorticoid secretion in PMAH patients per cell basis is reduced. Thus, ARMC5 KO mice are the first available animal model for PMAH, and we believe the scientific community will be very interested in this model for further mechanistic studies on BMAH. Therefore, timely publication of this work will benefit not only the communities of immunologists, but also endocrinologists. More detailed studies on the function of ARMC5 in the KO adrenal gland are in progress, but will be reported separately. Our current work remains focused on T cell immune responses.

4. Since the number of panels and figures in the paper are quite high, Fig 3d-f, Figure 4a-d and Fig 6a can be moved to a Supplemental as the findings are very similar in WT and KO.

As suggested by the reviewer, the original Fig. 3d and Fig. 4 is moved to the supplementary materials. Fig. 6a is an essential quality check of fetal liver transplantation, and is an integral part of the figure. So it remains in its original place.

5. Fig 7e label I in IL-17 is missing.

The missing label is added.

6. The concept figure (Fig.9) has no legend - this should be provided. The mechanism of action described in the Discussion can be improved and focused. As it stands it is rather diffuse.

A legend to figure 9 is added, and we emphasize that this model is speculative but demonstrates a principle of ARMC5 action mechanisms. We have now mentioned that the exact ARMC5-binding molecules in the graph are subjected to change upon data from further confirmatory studies. We have revised the discussion to make it more focused.

7. Spelling

a. Remove etc in introduction

b. Misspelling in last paragraph on page 4 that instead of than

These have been corrected.

Reviewer 3

In this manuscript, Hu et al. explore potential roles for the ARM domain-containing protein, Armc5. Through the use of a knockout strategy in combination with descriptive and functional assays, the authors find that mice lacking Armc5 have severe developmental defects and ineffective immune responses. In particular, the authors make the novel finding that T cell responses, both CD4+ and CD8+, appear compromised, as these cells display defects in both proliferation and effector functions. Interestingly, T

cell populations in knockout mice are neither sufficient to support the complete onset of the autoimmune disease EAE, nor able to efficiently clear an acute LCMV infection. Finally, the authors utilize a yeast two-hybrid approach in an attempt to gain insight into the Armc5-dependent mechanisms that may be responsible for the observed defects. Overall, the authors present some intriguing and novel findings. However, the manuscript is largely descriptive and the absence of mechanistic data diminishes the significance and impact of the work in its current form. I have the following comments and/or concerns.

1. An immunoblot should be performed to examine Armc5 protein in wild-type vs. Armc5^{-/-} tissues. The data from Figure 2d are unimpressive as there is little difference between WT and KO. As such, it is particularly important that the antibody used for flow cytometry be validated by immunoblot. Also, it is difficult to assess the flow cytometry data without an isotype control being displayed. The use of an isotype control is critical. It should be shown.

We agree that the FACS staining of the KO and WT T cells with anti-ARMC5 Ab is not satisfactory. None of the commercial Abs (we tested all of them) can detect specifically the ARMC5 protein by Western or flow cytometry. The previous publications by other groups to measure ARMC5 proteins by Western are likely reporting non-specific bands. Our own Abs could not be used for Western, and when it is used in flow cytometry, it is slightly better than the commercial Ab, but as reviewers pointed out, the result is not very convincing. In view of such technical limitations, we decided to remove our original flow cytometry histogram and rely on RT-qPCR and Southern to prove the deletion of ARMC5 in the KO mice. Based on RT-qPCR and Southern, as well as the prominent phenotypes in the body size, T cells and adrenal glands of the KO mice, we hope reviewers agree that there is little doubt that ARMC5 is deleted.

In short, due to the limitation of the availability of specific anti-ARMC5 Abs, it is not possible to measure ARMC5 proteins by any means at the present time. There is no way of predicting when a good Ab, either made by us or by companies, will become available. We hope the reviewer appreciate this insurmountable technical difficulty.

2. The defects in T cell proliferation in Figures 5a and 5b are quite striking. However, it appears that only one cell division has been measured. A more extensive analysis may be interesting in that it would allow the authors to determine whether the block is complete or whether there is a more subtle delay in cell proliferation. A more detailed cell cycle analysis may also provide further information as to the stage of division in which the block occurs.

We have carried out additional experiments to show the compromised proliferation of the KO T cells. Collectively, they are not totally blocked in cycling but proliferate slower than the WT T cells. We have carried out cell cycle analysis *in vitro*, and showed that KO T cells are compromised in G1/S-phase entry (Fig. 4b). As primary spleen T cells are difficult to be blocked by nacodazole and then released at the G2/M phase, we could not test whether ARMC5 has a function in G2/M phase entry. However, most cell cycle regulation events occur at the G1/S phase boundary.

3. The authors alternate between using feeder cells in combination with anti-CD3 and anti-CD3/anti-CD28 in a number of experiments (Figs. 5a, 5b, 5c, 6b, 6c). It is difficult to discern why the authors chose to do this and which experiments were performed with each specific set of conditions. Were differences observed in experimental outcome depending on the particular method of cell stimulation? The authors should clearly state why alternating conditions were used and the condition that was chosen for a particular experiment.

For *in vitro* differentiation of Th0, Th1 and Th17 cells, we always use feeder cells plus anti-CD3. This gives best results. The data presented in Fig. 5 and 6 are based on such stimulation. For cell proliferation, we tested different stimulations, such as CD3 plus CD28-coated wells for purified T cells, and Dynabeads coated with anti-CD3 and anti-CD28 for purified T cells in our previous studies, and total spleen cells stimulated with soluble CD3 and then gated on CD4 and CD8 cells. The last mentioned gives the best response, showing multiple rounds of proliferation according to CFSE labeling. Therefore, in the revised version, we have presented results based on the last mentioned way of stimulation. For RT-qPCR measurements of ARMC5 expression in CD4 and CD8 T cells during their activation and differentiation, we have to use Dynabeads coated with anti-CD3 and anti-CD28, so that we can get RNA from highly purified CD4 or CD8 cells in the absence of contaminating APC. The reasons are now stated in the section of methods.

4. As with the defects in proliferation, the T cell differentiation experiments are very intriguing. However, outside of the analysis of IFN γ and IL-17 expression, we know very little with regards to the mechanism resulting in the defect. Is the defect specific to cytokine (IFN γ , IL-17) secretion? What is the expression of T-bet and ROR gamma T? Do these cells express cytokine receptors consistent with Th1 and Th17 development? The impact of the current data would be significantly enhanced with an attempt to further elucidate the mechanism behind the developmental defect.

Our original data showed the reduced Th1 and Th17 cells among CD4 cells, which are already proliferated during the differentiation period (from 0.2×10^6 cells/well at the beginning to $0.5\text{-}2 \times 10^6$

cells /well at the end of differentiation). This means the cells are defective in differentiation. We have now analyzed the proliferation of these cells during Th1 and Th17 differentiation. These cells are defective in proliferation as well under these conditions as shown by CFSE staining (Fig. 5a). So both proliferation and differentiation under the Th1 and Th17 conditions are compromised in KO CD4 cells. We also provided new evidence that T-bet and ROR γ t expressions has no significant abnormality in KO Th1 and Th17 CD4 cells, respectively (Figs. 5c and 5d). Cytokine receptors important in for Th1 and Th17 expansion, such as IL-12R β 2 and IL-23R, are not reduced in KO CD4 cells cultured under the Th1 and Th17 conditions, respectively, compared to their WT counterparts (S. Figs. 5a and 5b).

5. What are the dynamics of Armc5 expression during T cell development? Outside of figure 1D, there is no attempt to examine Armc5 expression (particularly at the protein level) during T cell proliferation or activation. Expression patterns should be assessed as these data could provide insight into potential mechanistic roles for Armc5 in T cell differentiation.

We have now provided new data about ARMC5 expression during Th0, Th1, Th17 differentiation (S. Fig. 11), during CD8 cell activation (S. Fig. 13), and in thymocyte development, i.e., in SP, DP, DN1-4 thymocytes (S. Figs. 1b and 1c). Due to the unavailability of quality anti-ARMC5 Abs, as eluded above, these expression studies were performed using RT-qPCR. We hope the reviewer appreciate such a technical limitation.

6. There is not sufficient data to justify the model proposed in Figure 9.

We concur that the model is speculative at this time, but hope this diagram help the readers to visualize the general principle of how ARMC5 works. We have now moved this figure into the supplementary data, and emphasize in the legend that this model is speculative and the exact ARMC5-binding molecules in the graph subject to change upon data from further confirmatory studies.

Of course, if the reviewer insists to remove this figure, we will comply.

Minor concerns

1. No Th0 control for the Th17 cells is shown in Figure 6d.

Th0 controls are added to Th17 cells in Figure 6d.

2. *The assertion that the immune response to LCMV is largely CD8+ cell mediated is overstated. LCMV has been used extensively to study CD4+ T cell responses. As such, the CD4+ response in the LCMV studies (Figure 8) may be an area of interest for the authors to explore.*

CD4 cells must have some functions in anti-LCMV immune responses, although they are not the dominantly expanded population. In any case, we have now rephrased the importance regarding CD8 cells, and provided results regarding the CD4 cell expansion during LCMV infection (Fig. 8a) and LCMV-specific cytokine production by CD4 cells (Fig. 8f).

3. *The image in Figure 1e is very dark and thus may be difficult for the audience to visualize. Also, as this is human ARMC5, does murine Armc5 similarly display cytosolic localization?*

We have now improved the contrast of the image. It is largely due to display of the image on the screen. There was an error in the method section. The plasmid actually expresses mouse ARMC5 but not human ARMC5. We thank the reviewer for point out this error.

A note to all reviewers: We have added an interesting figure (S. Fig. 9f) which shows increased dysfunctional PD-1⁺/Tim3⁺ CD8 T cells in the KO spleen during LCMV infection. This experiment is not required by any reviewers. Presently, this experiment is done in 2 KO and 3 WT mice. Due to logistic limitations (a. we have used up all our KO mice for the revision. b. KO mice occurs below the Mendalian ratio, and it takes a long time for us to breed KO mice. c. Long-term quarantine is needed during the transferring of the KO mice to our collaborator for LCMV studies), adding a few more mice to this experiment might take a year. We propose to keep this figure in the supplementary data. However, if the reviewers consider the data not reliable due n=2 for the KO mice, we could remove this figure and the mentioning of it in the text.

Reviewers' comments:

Reviewer #1 (Remarks to the Author):

In this revised version, Hu et al have significantly improved their manuscript. Based on their analysis it appears clear that ARMC5 plays an important role in T cell proliferation and differentiation.

There are however a few points that should be addressed before publication.

1. The authors should show some data related to the activation of the T cells in vitro. Do these cells get properly activated, ie do they up-regulate molecules such as CD25 or CD69?
2. The authors show some basic Treg analysis in their response to the reviewer. These should be included in the manuscript. This will not impact on a future more detailed analysis of Tregs.
3. The B cell and Tfh analysis provided is very poor and not informative. The authors should provide some basic flow-cytometry data showing proportions of CD19+ B cells and maturation status of B cells. The current figures S6, S7 and S14 are not informative and should be deleted.
4. Figure 6a should be moved to Suppl. Data.
5. The analysis in Figure 6 looks odd. What are the cells that are not CD4+? The percentages shown in this figure should reflect the proportion of cytokine producing CD4 T cells (not the % of everything).
6. I don't think Suppl. Figure 5 is correct or informative. I suggest cutting it.
7. Suppl. Fig. 9 is important and I suggest moving some of the data to the primary figures, for example panel a and b. However, in its current form that is a problem with panel b – I suspect that IL7R and KLRG1 were confused. Panel c is uninformative since there are hardly any events in the plots. The authors should use total CD8 here. Panel f is too preliminary and should be removed.
8. The images in figure still need to be improved.
9. In multiple places throughout the manuscript, the authors refer to 'histograms' but really are talking about 'flow-cytometry plots' or 'dot plots'.

Reviewer #2 (Remarks to the Author):

The comments I made to the original manuscript have been answered. A technical detail. The Supplement showed have Figures and corresponding legends displayed together for readability

Reviewer #3 (Remarks to the Author):

In this revised manuscript, Hu et al. have added a number of pieces of new data that enhance and support the findings from their original manuscript. In particular, new data further exploring the proliferative defects observed in Armc5 knockout cells, as well as data further defining defects in Th1, Th17, and CD8 populations are significant additions. While the manuscript still largely lacks any mechanistic insight into the function of Armc5, the described defects in T cell development and function are of interest.

I have the following remaining concerns and/or suggestions:

1. In the absence of additional supporting mechanistic data, this Reviewer is still in favor of eliminating Supplementary Figure 15.
2. Figure 1e is still very difficult to visualize.
3. Figure 5a (right panel) is mislabeled. I believe "IFN-g" should read "IL-17".
4. Figure 4d is currently represented as fold relative expression. The data may be more insightful if represented as relative expression (similar to that in Figure 2c).
5. Figure 5b. Labels indicating the day of observation would be useful.

Point-to-point answers to reviewers' comments.

The original comments are in italic.

Reviewer #1

In this revised version, Hu et al have significantly improved their manuscript. Based on their analysis it appears clear that *ARMC5* plays an important role in T cell proliferation and differentiation.

There are however a few points that should be addressed before publication.

1. The authors should show some data related to the activation of the T cells in vitro. Do these cells get properly activated, ie do they up-regulate molecules such as CD25 or CD69?

We tested these activation makers. The T cells under our experimental condition are well activated with drastic CD25 and CD69 upregulation. No difference between KO and WT is observed. These negative data are presented below for the reviewers and are mentioned in the text as "data not shown".

Figure 1 for reviewers. KO T cells have normal CD25 and CD69 upregulation after TCR activation

Spleen T cells from WT and KO mice were cultured in the absence or presence of anti-CD3ε Ab (1 µg/ml) for 16 h. CD4 (left panels) and CD8 (right panels) were gated and analyzed for their CD25 and CD69 expression. The experiment was repeated more than 3 times, and representative dot plots are shown.

2. The authors show some basic Treg analysis in their response to the reviewer. These should be included in the manuscript. This will not impact on a future more detailed analysis of Tregs.

We have now move the 2 figures related to Treg from “for reviewers” to the formal text, as suggested.

3. The B cell and Tfh analysis provided is very poor and not informative. The authors should provide some basic flow-cytometry data showing proportions of CD19+ B cells and maturation status of B cells. The current figures S6, S7 and S14 are not informative and should be deleted.

As per reviewer’s suggestion, S6, S7 and S14 are deleted”.

The reviewer wants to see the B cell maturation markers. It is not a difficult experiment to do. Had this request been raised in the last review, we could easily include the analysis in our previous experiments.

Now, we have performed a preliminary analysis of plasma cells in the periphery and early B cells in the bone marrow, using one pair of KO and WT mice. The data are shown below in figures 2 and 3 for reviewers. No apparent difference between WT and KO mice in their B cell maturation is observed.

However, we are not able to conduct a confirmatory and comprehensive analysis on B cells, because we have run out of the supply of KO mice, which were all used up to address various critiques in the last round of review. The KO mice are difficult to breed, and at our current rate, it will take 3 to 5 months to get enough KO mice to repeat this B cell maturation experiment, which is really a minor issue and is not in the scope of this study, as the title of our paper is “ARMC5 deletion causes developmental defects and compromises **T-cell immune responses**”. The reviewer acknowledged in his previous review that “I understand that not everything can be analyzed in the first report” and raised his request on B cell responses as a minor point. We will still include serum IgG levels and B cell proliferation of KO and WT mice, to globally answer the question whether there is major defect in the B cell compartment. However, we hope the reviewer agrees with us that in the interest of timely publication of this work, it will be a pity to wait for 5 months to add this minor information about B cell maturation markers, which will be negative anyways as shown in the figures below.

Therefore, we are seeking reviewer 1’s consent not to present data related to B cells, with the exception of total blood IgG levels and B cell proliferation.

Figure 2 for reviewers. No apparent difference in plasma cell differentiation in WT and KO mice

LN cells from WT and KO mice were analysis for their B220 and CD138 expression by flow cytometry, and dot plots are shown. There is no apparent difference in the percentages of $B220^{int}CD138^{+}$ plasma blasts, $B220^{lo}CD138^{+}$ plasma cells, and $B220^{+}CD138^{-}$ B cells in LN from WT versus KO mice.

Figure 3 for reviewers. No apparent difference in early B cell differentiation in WT and KO mice

Bone marrow cells from WT and KO mice were gated on B220⁺ cells and were further analyzed for their expression of B cell maturation makers by flow cytometry. Gating strategy and percentages of different populations are shown in the dot plots.

Upper row: Fractions A-C represent pro-B cells

Lower row: Fraction D represents pre-B cells; fraction E represents immature B cells; fraction F contains mature recirculating B cells. (Hardy et al. J. Exp. Med. 173:1213, 1991).

No apparent difference is observed with regard to the percentages of pro-B, pre-B and immature B cells in the bone marrow of WT versus KO mice.

4. Figure 6a should be moved to supplementary data.

It is done.

5. The analysis in Figure 6 looks odd. What are the cells that are not CD4+? The percentages shown in this figure should reflect the proportion of cytokine producing CD4 T cells (not the % of everything).

Those were feeder cells. They are now gated out and percentages are recalculated. We thank the reviewer for pointing out this error in presentation.

6. I don't think Suppl. Figure 5 is correct or informative. I suggest cutting it.

This figure is deleted.

7. Suppl. Fig. 9 is important and I suggest moving some of the data to the primary figures, for example panel a and b. However, in its current form that is a problem with panel b – I suspect that IL7R and KLRG1 were confused. Panel c is uninformative since there are hardly any events in the plots. The authors should use total CD8 here. Panel f is too preliminary and should be removed.

S. Fig. 9a and 9b are moved to the text. Indeed, axes of panel B were mislabeled. We thank the reviewer for pointing this out and it is now corrected. For panel C (now S. Fig. 7A), the low events in KO spleen were due to low number of tetramer-positive cells after the clonal expansion, and this represent a technical difficulty to obtain enough events. However, our bar graphs summarizing these data are based on results from 5-8 mice, and are therefore reliable. We have kept the original dot plot in S. Fig 7A (right panel), but as per reviewer's instruction, added a panel showing results on CD8-gated cells for their CD62L and CD44 expression (S. Fig. 7A, right panel). We can get comparable event in WT and KO cells by gating on CD8 cells, although the cells are not all LCMV-specific.

8. The images in figure still need to be improved.

We are now using better micrographs for Fig. 1E.

9. In multiple places throughout the manuscript, the authors refer to 'histograms' but really are talking about 'flow-cytometry plots' or 'dot plots'.

This has been corrected.

Reviewer 2:

The comments I made to the original manuscript have been answered. A technical detail: the Supplement showed have Figures and corresponding legends displayed together for readability.

The style has been changed as suggested by the reviewers.

Reviewer #3:

In this revised manuscript, Hu et al. have added a number of pieces of new data that enhance and support the findings from their original manuscript. In particular, new data further exploring the proliferative defects observed in Armc5 knockout cells, as well as data further defining defects in Th1, Th17, and CD8 populations are significant additions. While the manuscript still largely lacks any mechanistic insight into the function of Armc5, the described defects in T cell development and function are of interest.

I have the following remaining concerns and/or suggestions:

1. In the absence of additional supporting mechanistic data, this Reviewer is still in favor of eliminating Supplementary Figure 15.

This figure is deleted.

2. Figure 1e is still very difficult to visualize.

A new and better Fig. 1E is used now.

3. Figure 5a (right panel) is mislabeled. I believe "IFN-g" should read "IL-17".

The miss-labelling is corrected.

4. Figure 4d is currently represented as fold relative expression. The data may be more insightful if represented as relative expression (similar to that in Figure 2c).

There is no Figure 4d in our manuscript. Actually, none of the d panels in any figure fits the said description.

5. Figure 5b. Labels indicating the day of observation would be useful.

The days of observation are added now to Fig. 5b.